# Faithful Interpretation for Graph Neural Networks

**Lijie Hu**[*]                                                    *lijie.hu@kaust.edu.sa*
*Provable Responsible AI and Data Analytics (PRADA) Lab*
*King Abdullah University of Science and Technology*

**Tianhao Huang**[*]                                              *tianhao.alex.huang@gmail.com*
*Provable Responsible AI and Data Analytics (PRADA) Lab*
*Nankai University*

**Lu Yu**                                                          *lu.yu@kaust.edu.sa*
*Ant Group*

**Wanyu Lin**                                                      *wanylin@comp.polyu.edu.hk*
*Department of Computing*
*Hong Kong Polytechnic University*

**Tianhang Zheng**                                                *zthzheng@zju.edu.cn*
*The State Key Laboratory of Blockchain and Data Security*
*Zhejiang University*

**Di Wang**[†]                                                     *di.wang@kaust.edu.sa*
*Provable Responsible AI and Data Analytics (PRADA) Lab*
*King Abdullah University of Science and Technology*

**Reviewed on OpenReview:** *https://openreview.net/forum?id=Y8EspxaksH*

## Abstract

Currently, attention mechanisms have garnered increasing attention in Graph Neural Networks (GNNs), such as Graph Attention Networks (GATs) and Graph Transformers (GTs). This is due to not only the commendable boost in performance they offer but also their capacity to provide a more lucid rationale for model behaviors, which are often viewed as inscrutable. However, Attention-based GNNs have demonstrated instability in interpretability when subjected to various sources of perturbations during both training and testing phases, including factors like additional edges or nodes. In this paper, we propose a solution to this problem by introducing a novel notion called Faithful Graph Attention-based Interpretation (FGAI). In particular, FGAI has four crucial properties in terms of stability and sensitivity to interpretation and the final output distribution. Built upon this notion, we propose an efficient methodology for obtaining FGAI, which can be viewed as an ad hoc modification to the canonical Attention-based GNNs. To validate our proposed solution, we introduce two novel metrics tailored for graph interpretation assessment. Experimental results demonstrate that FGAI exhibits superior stability and preserves the interpretability of attention under various forms of perturbations and randomness, which makes FGAI a more faithful and reliable explanation tool.

## 1 Introduction

Graph Neural Networks (GNNs) have experienced rapid proliferation, finding versatile applications across a spectrum of domains such as communication networks (Jiang, 2022), medical diagnosis (Ahmedt-Aristizabal

---

[*]Equal contribution. Part of the work was done when Tianhao Huang was a research intern at PRADA Lab.
[†]Corresponding author.

et al., 2021), and bioinformatics (Yi et al., 2022). As the adoption of deep neural networks continues to expand within these diverse fields, the significance of interpreting deep models to improve decision-making and establish trust in their results has grown commensurately. In this context, Attention-based GNNs, such as Graph Attention Network (GAT) (Veličković et al., 2018) and Graph Transformer (GT) (Dwivedi & Bresson, 2021), have arisen as a prominently utilized approach for model interpretation, which offers the promise of deciphering complex relationships within graph-structured data using attention vectors, providing valuable insights into the decision-making processes of deep models.

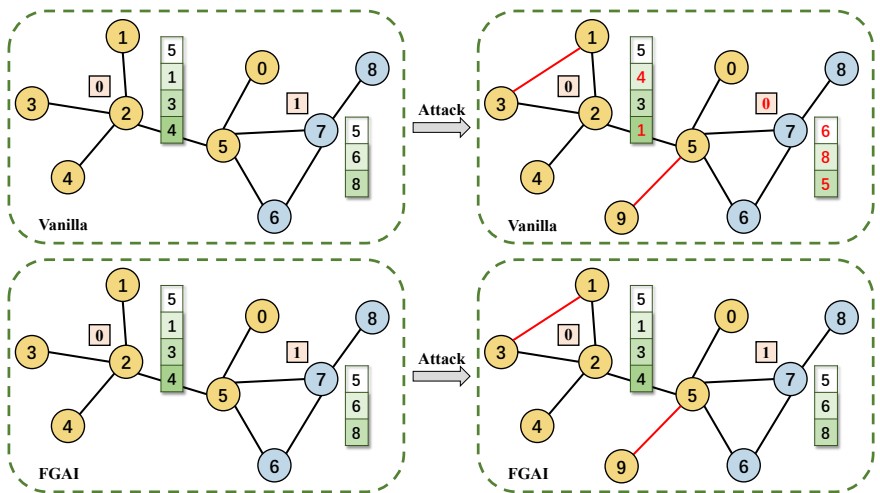

Figure 1: Stability of interpretation and prediction in a binary node classification task. Interpretation and predictions within the binary node classification can be susceptible to perturbations.

However, the maturation of GNNs has revealed a critical concern that the increasing prevalence of adversarial attacks targeting graph-structured data. Dai et al. (2018); Zügner et al. (2018; 2020) has also exposed potential vulnerabilities in Attention-based GNNs. This necessitates a meticulous examination of Attention-based GNNs' robustness and reliability of interpretation in real-world applications. As illustrated in Figure 1, our investigations have uncovered that even slight perturbations, such as the addition of edges to a node and the introduction of a new node, can significantly change the predictions and interpretations of GAT. Notably, these perturbations lead to shifts in top indices of the attention vector, resulting in significant alterations to the prediction for node 7. Recent studies (Li et al., 2024; 2025) have also shown that even minor perturbations to graph structure can significantly alter the explanation outcomes of Explainable Graph Neural Networks (XGNNs), posing serious risks in safety-critical applications such as drug discovery. This vulnerability challenges the fundamental principle that slight perturbations should not drastically alter attention distributions, as unstable explanations undermine trust in model interpretability. This inherent instability fundamentally undermines Attention-based GNNs' capacity to serve as a truly faithful tool for model explanation.

Instability has been a recognized problem in interpretation methods within deep learning. An unstable interpretation is susceptible to noise in data, hindering users from comprehending the underlying logic of model predictions. Furthermore, instability reduces the reliability of interpretation as a diagnostic tool, where even slight input perturbations can drastically alter interpretation outcomes (Ghorbani et al., 2019; Dombrowski et al., 2019; Yeh et al., 2019). For safety- and security-critical applications like drug discovery, slight perturbations may have disastrous effects on the results. For example, Wang et al. (2023) introduces DrugExplorer to provide domain experts with interpretable visual explanations for GNN-based drug repurposing predictions. Hence, stability is increasingly recognized as a crucial factor for faithful model interpretations. This prompts our question: *Can we augment the faithfulness of attention layers by enhancing their stability while preserving the key characteristics of intermediate representation explanation and prediction?*

To advance the cause of faithful graph interpretation within the graph attention-based framework, we introduce a rigorous definition of Faithful Graph Attention-based Interpretation (FGAI) that operates at node classification. Attention vectors in the graph attention-based networks provide visual representations of the importance weights assigned to neighboring nodes. Thus, intuitively, FGAI should possess the following four properties for any input: (1) Significant overlap between the top-$k$ indices of the attention layer of FGAI and the vanilla attention to inherit interpretability from the original Attention-based GNNs. (2) Inherent stability, making the attention vector robust to randomness and perturbations during training and testing. (3) Close prediction distribution to that of vanilla Attention-based GNNs to maintain Attention-based GNNs' outstanding performance. (4) Stable output distribution, which is robust to randomness and perturbations during training and testing. Based on these four criteria, we can formally define the FGAI as a substitute for attention-based graph interpretation.

Our contributions can be summarized as follows: (1) *In-depth analysis of stability in graph attention-based networks interpretation and prediction.* Our work explores and explains the stability issue of the attention layer in Attention-based GNNs caused by different factors. (2) *A Definition for FGAI.* We propose a rigorous mathematical definition of FGAI. This definition is for node classification, and it is also a formal framework for faithful interpretation in Attention-based GNNs. (3) *An Efficient algorithm for FGAI.* To effectively address the stability issue identified and aligned with our proposed definition of FGAI, we introduce a novel method for deriving FGAI. Specifically, we present a minimax stochastic optimization problem designed to optimize an objective function comprising four key terms, each corresponding to one of the four essential properties outlined in our definition. (4) *Novel evaluation metrics.* We design two innovative evaluation metrics for rigorously evaluating the attention-based graph interpretation techniques. The comprehensive experiments showcase the superior performance of our approach in enhancing the faithfulness of Attention-based GNN variants, indicating that FGAI is a more reliable and faithful explanatory tool. Due to the space limit, we provide additional experimental settings and results in the appendix.

## 2 Related Work

**Robustness in Graph Neural Networks.** There has been some research investigating the robustness of graph-based deep learning models. Verma & Zhang (2019) first derived generalization bounds for GNNs and explored stability bounds on semi-supervised graph learning. Zügner & Günnemann (2019) and Bojchevski & Günnemann (2019) independently developed effective methods for certifying the robustness of GNNs, and they also separately introduced novel robust training algorithms for GNNs. However, to the best of our knowledge, there has been limited research conducted on the robustness of GAT. Brody et al. (2021) indeed pointed out that their proposed GATv2 has an advantage over the vanilla GAT in terms of robustness. Nevertheless, their approach sacrifices the interpretability that is present in the original attention. GSAT (Miao et al., 2022) leverages stochastic attention and information bottleneck to achieve interpretable and generalizable graph learning, surpassing both post-hoc methods and existing interpretable models without sacrificing predictive performance. However, it does not address the issue of how to maintain the robustness of explanations under graph perturbations.

**Faithfulness in Explainable Methods.** Faithfulness is an essential property that explanation models should satisfy, which refers to the explanation should accurately reflect the true reasoning process of the model (Wiegreffe & Pinter, 2019; Herman, 2017; Jacovi & Goldberg, 2020; Lyu et al., 2022; Hu et al., 2024a;b;c). Faithfulness is also related to other principles like sensitivity, implementation invariance, input invariance, and completeness (Yeh et al., 2019). Completeness means that an explanation should comprehensively cover all relevant factors for prediction (Sundararajan et al., 2017). The other three terms are all related to the stability when facing different kinds of perturbations. The explanation should change if heavily perturbing the important features that influence the prediction (Adebayo et al., 2018), but stable to small perturbations (Hu et al., 2023b;a; Gou et al., 2023; Lai et al., 2023). Thus, stability is an important factor in explanation faithfulness. Some preliminary work has been proposed to obtain stable interpretations. For example, Yeh et al. (2019) theoretically analyzed the stability of post-hoc interpretation and proposed to use smoothing to improve interpretation stability. Yin et al. (2022) designed an iterative gradient descent algorithm to get counterfactual interpretation, which shows desirable stability. Chen et al. (2024) theoretically analyze the

limitations of attention-based interpretable GNNs via subgraph multilinear extension, and propose GMT to bridge the approximation gap, achieving superior interpretability and generalizability. While some prior work has proposed techniques to attain stable interpretations, these approaches have primarily focused on post-hoc interpretation and text data, making them less directly applicable to the complexities of graph data. In this context, our work bridges these research areas by addressing the stability problem in graph interpretation, providing a rigorous definition of Faithful Graph Attention-based Interpretation (FGAI), proposing efficient methods aligned with this definition, and introducing novel evaluation metrics tailored to the unique demands of graph interpretation, thereby advancing the faithfulness and robustness of graph-based deep learning models.

**Attention-based Graph Neural Networks.** Graph Neural Networks (GNNs) (Scarselli et al., 2008; Gori et al., 2005) have garnered significant attention in recent years as a deep learning approach in the fields of graph data mining and machine learning. Prior studies have put forth a variety of GNN variants featuring diverse approaches to aggregating neighborhood information and employing graph-level pooling methods, such as GraphSAGE (Hamilton et al., 2017), Graph Convolutional Network (GCN, (Kipf & Welling, 2016)), Graph Attention Network, (GAT, (Veličković et al., 2018)), Graph Isomorphism Network, (GIN, (Xu et al., 2018)), and others. Among them, GAT (Veličković et al., 2018) introduced an attention mechanism during the node aggregation process, enabling the network to learn the importance weights between different nodes automatically. This capability allows GAT to better handle heterogeneity and importance differences among nodes, making it one of the state-of-the-art graph neural networks and exhibiting strong interpretability. Wang et al. (2019) introduced an innovative heterogeneous graph neural network that leverages hierarchical attention to achieve an optimal fusion of neighbors and multiple meta-paths in a structured manner. Brody et al. (2021) revealed that vanilla GAT computes limited attention that the ranking of the calculated attention scores remains unconditioned and proposed a GATv2 with a modified order of operations that demonstrates superior performance compared to GAT, exhibiting enhanced robustness to noise as well.

## 3 Toward Faithful Graph Attention-based Interpretation

### 3.1 Graph Attention Networks

Before clarifying the definition of the model with stable interpretability, we first introduce the attention layer in Attention-based GNNs. Note that our approach can be directly applied to any graph attention-based network. Here, we follow the notations in Veličković et al. (2018) for typical GAT. We have input nodes with features $h = \{h_1, h_2, \ldots, h_N\}$, $h_i \in \mathbb{R}^F$, where $N$ is the number of nodes and $F$ is the dimension of features. The input node features are fed into the GAT layer and then produce a new set of node features $h' = \{h'_1, h'_2, \ldots, h'_N\}$, $h'_i \in \mathbb{R}^{F'}$. Then, the attention coefficients are calculated by a self-attention mechanism on the nodes: $a = \mathbb{R}^{F'} \times \mathbb{R}^{F'} \to \mathbb{R}$

$$e_{ij} = a(\boldsymbol{W}h'_i, \boldsymbol{W}h'_j), \tag{1}$$

where $\boldsymbol{W} \in \mathbb{R}^{F'} \times \mathbb{R}^{F'}$ is a weight matrix after a shared linear transformation applied to each node. And $e_{ij}$ indicates the importance of node $j$'s features to node $i$. In practice, a softmax function is used to normalize the coefficients across all neighbors of each node. Specifically, for node $i$ we have $w_i \in \mathbb{R}^{|\mathcal{N}_i|}$ with each entry $w_{ij}$ being

$$w_{ij} = \text{softmax}_j(e_{ij}) = \frac{\exp(e_{ij})}{\sum_{k \in \mathcal{N}_i} \exp(e_{ik})}, \tag{2}$$

where $\mathcal{N}_i$ is the neighbor set of node $i$. Subsequently, $w_{ij}$ are employed to calculate a linear combination of the corresponding features, serving as the final output features for each node. We consider the node classification task. The final output distribution for each node is

$$y_i = \sigma\left(\sum_{j \in \mathcal{N}_i} w_{ij} \boldsymbol{W} h_j\right), \tag{3}$$

where $\sigma$ is a non-linear activation function.

### 3.2 Faithful Graph Interpretation

**Motivation:** As we discussed in the introduction section, for Attention-based GNNs, both the interpretability of the attention vector and the performance of the classifier are unstable to slight perturbations on the graph structure, invalidating it as a faithful explanation tool. Thus, we aim to find a variant of the attention vector that is stable while preserving the key characteristics of intermediate representation explanation and prediction in vanilla attention. We call such a variant "stable attention". Before diving into our rigorous definition of "stable attention", we first need to intuitively think about what properties it should have. The first one is keeping similar interpretability as the vanilla attention. In the vanilla attention vector for a node, we can easily see that the rank of its entries can reflect the importance of its neighboring nodes. Thus, "stable attention" should also have almost the same order for each entry as in the vanilla one. However, keeping the rank for all entries is too stringent, motivated by the fact that the interpretability and the prediction always rely on the most important entries. Here, we can relax the requirement to keep the top-$k$ indices almost unchanged.

In fact, such a property is not enough as it could also be unstable. Modeling such instability is challenging as the perturbation could be caused by multiple resources, which is significantly different from adversarial robustness. Our key observation is that wherever a perturbation comes from, it will subsequently change the attention vector. Thus, if the interpretability of "stable attention", i.e., the top-$k$ indices, is resilient to noise, we can naturally think it is robust to those different perturbations.

However, this is still insufficient. The main reason is that keeping interpretability does not indicate keeping the prediction performance. This is because keeping interpretability can only guarantee the rank of indices unchanged, but cannot ensure the magnitude of these entries, which determine the prediction, unchanged. For example, suppose the vanilla attention vector is $(0.5, 0.3, 0.2)$, then the above "stable attention" might be $(0.9, 0.051, 0.049)$, which is significantly different from the original one. Based on these, we should also enforce the prediction performance, i.e., the output distribution, to be almost the same as the vanilla one. Moreover, its output distribution should also be robust to perturbations shown in Figure 4 in Appendix.

Based on our above discussion, our takeaway is that "stable attention" should make its top-$k$ indices and output distribution almost the same as vanilla attention while also being robust to perturbations. In the following, we will translate the previous intuitions into rigorous mathematical definitions. Specifically, we call the above "stable attention" as Faithful Graph Attention-based Interpretation (FGAI). We first give the definition of top-$k$ overlaps to measure the interpretability stability.

**Definition 3.1** (**Top-$k$ overlaps**). For vector $x \in \mathbb{R}^d$, we define the set of top-$k$ component $T_k(\cdot)$ as follow,

$$T_k(x) = \{i : i \in [d] \text{ and } \{|\{x_j \geq x_i : j \in [d]\}| \leq k\}\}.$$

And for two vectors $x$, $x'$, the top-$k$ overlap function $V_k(x, x')$ is defined by the overlapping ratio between the top-$k$ components of two vectors,[*] i.e., $V_k(x, x') = \frac{1}{k}|T_k(x) \cap T_k(x')|$.

Moreover, from equation 1 and equation 2, we can see the vanilla attention vector $w_i$ for node $i$ depends on input $h_i$, thus we can think of it as a function of $h_i$ denoted as $w(h_i)$. Similarly, the output distribution $y_i$ is a function of an attention vector $w$ and the input node $h_i$, denoted as $y(h_i, w)$. Based on the above notation, we can formally define a (node-level) FGAI as follows.

**Definition 3.2** (**(Node-level) FGAI**). We call a map $\tilde{w}$ is a $(D, R, \alpha, \beta, k_1, k_2)$-Faithful Graph Attention-based Interpretation mechanism for the vanilla attention mechanism $w$ if it satisfies for any node $i$ with feature $h_i$,

- (**Similarity of Interpretability**) $V_{k_1}(\tilde{w}(h_i), w(h_i)) \geq \beta_1$ for some $1 \geq \beta_1 \geq 0$.

- (**Stability of Interpretability**) $V_{k_2}(\tilde{w}(h_i), \tilde{w}(h_i) + \rho) \geq \beta_2$ for some $1 \geq \beta_2 \geq 0$ and all $\|\rho\| \leq R_1$, where $\|\cdot\|$ is a norm and $R_1 \geq 0$.

- (**Closeness of Prediction**) $D(y(h_i, \tilde{w}), y(h_i, w)) \leq \alpha_1$ for some $\alpha_1 \geq 0$, where $D$ is some loss or divergence, $y(h_i, \tilde{w}) = \sigma(\sum_{j \in \mathcal{N}_i} (\tilde{w}(h_i))_j \boldsymbol{W} h_j)$.

---

[*]It is notable that the dimensions of $x$, $x'$ in the above definition could be different.

- **(Stability of Prediction)** $D(y(h_i, \tilde{w}), y(h_i, \tilde{w} + \delta)) \leq \alpha_2$ for all $\|\delta\| \leq R_2$, where $D$ is some loss or divergence, and $y(h, \tilde{w} + \delta) = \sigma(\sum_{j \in \mathcal{N}_i} (\tilde{w}(h_i) + \delta)_j \boldsymbol{W} h_j)$, $R_2 \geq 0$,

where $\alpha = \min\{\alpha_1, \alpha_2\}$, $\beta = \max\{\beta_1, \beta_2\}$, and $R = \min\{R_1, R_2\}$.

Moreover, for any input node with feature $h$, we call $\tilde{w}(h)$ as a $(D, R, \alpha, \beta, k_1, k_2)$-Faithful Graph Attention-based Interpretation for this node.

Note that in the previous definition, there are several parameters. There are two properties - similarity and stability for prediction and interpretability, respectively.

The first two conditions are the similarity and stability of the interpretability. We ensure $\tilde{w}$ has similar interpretability with the vanilla attention mechanism. There are two parameters, $k_1$ and $\beta_1$. $k_1$ could be considered prior knowledge, i.e., we believe the top-$k_1$ indices of attention will play the most important role in making the prediction, or their corresponding $k_1$ features can almost determine its prediction. $\beta_1$ measures how much interpretability does $\tilde{w}$ inherit from vanilla attention. When $\beta_1 = 1$, then this means the top-$k_1$ order of the entries in $\tilde{w}$ is the same as it is in vanilla attention. Thus, $\beta_1$ should close to 1. The term stability involves two parameters, $R_1$ and $\beta_2$, which correspond to the robust region and the level of stability, respectively. Ideally, if $\tilde{w}$ satisfies this condition with $R_1 = \infty$ and $\beta_2 = 1$, then $\tilde{w}$ will be extremely stable w.r.t any randomness or perturbations. Thus, in practice, we wish $R_1$ to be as large as possible and $\beta_2$ to be close enough to 1.

The last two conditions are the similarity and stability of prediction based on attention. In the third condition, $\alpha_1$ measures the closeness between the prediction distribution based on $\tilde{w}$ and the prediction distribution based on vanilla attention. When $\alpha_1 = 0$, then $\tilde{w} = w$. Therefore, we hope $\alpha_1$ to be as small as possible. It is also notable that $D$ is the loss to measure the closeness of two distributions, which could also be some divergence. Similarly, the term stability involves two parameters, $R_2$ and $\alpha_2$, which correspond to the robust region and the level of stability, respectively. Ideally, if $\tilde{w}$ satisfies this condition with $R_2 = \infty$ and $\alpha_2 = 0$, then $\tilde{w}$ will be extremely stable w.r.t any randomness or perturbations. Thus, in practice, we wish $R_2$ to be as large as possible and $\alpha_2$ to be sufficiently small.

Thus, based on these discussions, we can see Definition 3.2 is consistent with our above intuition on graph faithful attention, and it is reasonable and well-defined.

**More discussions on top-$k$ conditions.** In fact, the inclusion of a top-$k$ condition within the graph interpretation framework serves as a dual-purpose mechanism that significantly motivates our research efforts. Firstly, this condition plays a pivotal role in retaining the most salient characteristics of node information. By focusing on the top-$k$ elements, we ensure that the most critical aspects of the graph structure and relevance are preserved, allowing for a more focused and interpretable process. This selective retention of key features is particularly valuable in complex and large-scale graph data, where identifying the most influential nodes can lead to more meaningful insights and informed decision-making.

Secondly, the top-$k$ condition acts as a potent sparsity accelerator for graph computation. By narrowing down the attention scope to the most relevant nodes, we effectively reduce the computational burden associated with graph processing. In our extensive experiments, we empirically demonstrate that the computational cost incurred by our approach is well-contained, requiring no more than 150% of the GPU memory utilized by the vanilla GAT and GT. This efficiency gain not only ensures the scalability of our approach but also underscores its practical viability for real-world applications. In essence, our research is driven by the dual aspiration of enhancing the interpretability and computational efficiency of graph-based deep learning models. By integrating the top-$k$ condition, we strike a balance between retaining essential information and optimizing computational resources, thereby empowering graph interpretation with greater fidelity and scalability.

Here we need note that FGAI is different from adversarial robustness. See Appendix B for details.

## 4  Finding FGAI

In the last section, we presented a rigorous definition of faithful node-level graph interpretation. To find such an FGAI, we propose to formulate a minimax optimization problem that involves the four losses associated with the four conditions in Definition 3.2.

Based on Definition 3.2, we can see that a natural way to find a $\tilde{w}$ is to minimize the prediction closeness with the vanilla attention while also satisfying other three conditions:

$$\min_{\tilde{w}} \mathbb{E}_h[D(y(h, \tilde{w}), y(h, w))]$$
$$\text{s.t. } \mathbb{E}_h\big[\max_{\|\rho\| \le R} V_{k_2}(\tilde{w}(h), \tilde{w}(h) + \rho)\big] \ge \beta_2,$$
$$\mathbb{E}_h[V_{k_1}(\tilde{w}(h), w(h))] \ge \beta_1,$$
$$\mathbb{E}_h\big[\max_{\|\rho\| \le R} D(y(h, \tilde{w}), y(h, \tilde{w} + \delta))\big] \le \alpha_2.$$

However, the main challenge is that the top-$k$ overlap function is non-differential, which makes the optimization problem hard to train. Thus, we seek to design a surrogate loss $\mathcal{L}_k(\cdot)$ for $-V_k(\cdot)$, which can be used in training. The details are in the Appendix A.

**Final objective function and algorithm.**  By transforming each constraint to a regularization, we can finally get the following objective function. Details are in the Appendix A.

$$\min_{\tilde{w}} \mathbb{E}_h[D(y(h, \tilde{w}), y(h, w)) + \lambda_1 \mathcal{L}_{k_1}(w, \tilde{w}) + \lambda_2 \max_{\|\delta\| \le R} D(y(h, \tilde{w}), y(h, \tilde{w} + \delta)) + \lambda_3 \max_{\|\rho\| \le R} \mathcal{L}_{k_2}(\tilde{w}, \tilde{w} + \rho)],$$

where $\mathcal{L}_k(\cdot)$ is defined in (9). The second term top-$k$ is substituted by a surrogate loss, which is differentiable and practical to compute via backpropagation. This term guarantees the explainable information of the attention. The third term $D$ is a min-max optimization controlled by hyperparameter $\lambda_2$ in order to find the maximum tolerant perturbation to the attention layer, which affects the final prediction. The final term $\mathcal{L}_{k_2}$ is also a min-max optimization to find the maximum tolerant perturbation to the intrinsic explanation of the attention layer. In other words, we derive a robust region using this min-max strategy.

To solve the above minimax optimization problem, we propose Algorithm 1. See Appendix A for more details.

## 5  Experiments

### 5.1  Experimental Setup

**Datasets.**  In this study, we employ several datasets encompassing small to large-scale graphs to conduct an exhaustive comparison of our approach with the baseline methods. For the node classification task, we utilize Amazon CS and Amazon Photo (McAuley et al., 2015), Coauthor CS and Coauthor Physics (Shchur et al., 2018), ogbn-arXiv (Wang et al., 2020; Mikolov et al., 2013). For the link prediction task, we adopt Cora, Citeseer and Pubmed (Sen et al., 2008). Finally, for the graph classification task, we make use of the D&D (Debnath et al., 1991), MUTAG (Dobson & Doig, 2003) and Politifact (Dou et al., 2021) dataset. We present the statistics of the selected datasets in Table 3, 4 and 5. We also provide a detailed description of these datasets in Appendix C.

**Baselines.**  We employ three attention-based models, namely Graph Attention Network (GAT) (Veličković et al., 2018), GATv2 (Brody et al., 2021), and Graph Transformer (GT) (Dwivedi & Bresson, 2021) as base models for three downstream tasks, namely node classification, graph classification and link prediction task. Due to space constraints, the results of the link prediction task are presented in Appendix D. Following the work of Zheng et al. (2021), we compare our method with vanilla methods and two general defense techniques: layer normalization (LN) and adversarial training (AT). We refer readers to Appendix G for implementation details.

**Post-hoc vs Self-explained.** Here, we need to clarify the distinction between our approach and post-hoc learning frameworks such as PGExplainer (Luo et al., 2020), SubgraphX (Yuan et al., 2021), RC-Explainer (Wang et al., 2022), etc. As shown in the research by Kosan et al. (2023), these explainers often exhibit significant instability when facing perturbations. Our method, on the contrary, has a distinct advantage in this regard. In other words, our approach is specifically designed to address this issue.

**Evaluation Metrics.** For assessing the model's performance, we utilize the F1-score as a primary metric. To comprehensively assess the stability of the model when encountering various forms of perturbations and randomness, as well as its ability to maintain the interpretability of attention, we present graph-based Jensen-Shannon Divergence (g-JSD) and Total Variation Distance (g-TVD) as metrics for measuring model stability. Furthermore, we design two additional novel metrics, namely, $F_{slope}^+$ and $F_{slope}^-$, to fully evaluate the interpretability of the graph attention-based neural networks.

**i) Graph-based Total Variation Distance.** g-TVD is a metric employed to quantify the dissimilarity between two probability distributions. It is defined mathematically as

$$\text{g-TVD}(y, \tilde{y}) = \frac{1}{2|N|} \sum_{i=1}^{N} |y_i - \tilde{y}_i|,$$

where $y$ and $\tilde{y}$ represent the outputs of the model before and after perturbation of the graph, respectively. Note that compared to the original TVD, here we rescale it by dividing $|N|$, where $|N|$ is the number of nodes in the graph.

**ii) Graph-based Jensen-Shannon Divergence.** g-JSD is a metric used to quantify the similarity or dissimilarity between two probability distributions. It is defined as follows,

$$\text{g-JSD}(w, \tilde{w}) = \frac{1}{2|E|}(KL(w||\bar{w}) + KL(\tilde{w}||\bar{w})),$$

where $w$ and $\tilde{w}$ represent the attention vectors of the model before and after perturbation of the graph, $\bar{w} = \frac{1}{2}(w + \tilde{w})$, and $KL$ is Kullback-Leibler Divergence which can be defined as $KL(w||\bar{w}) = \sum_{i=1}^{E} w_i \log(\frac{w_i}{\bar{w}_i})$. Note that compared to the original JSD, here we rescale it by dividing $|E|$, where $|E|$ is the number of edges in the graph.

**iii) F-slope.** Building upon the foundation laid by Yuan et al. (2022), we propose $F_{slope}^+$ and $F_{slope}^-$ to better evaluate the interpretability of different Attention-based GNNs. In detail, let $T$ represent the set of nodes correctly classified by the model. Then, we rank the importance of edges based on the attention values assigned to each edge by the trained model. A higher attention value indicates greater importance, while a lower value suggests lesser importance for the respective edge on the graph. Next, we utilize $M_r^+$ as a mask for important edges and $M_r^-$ as a mask for unimportant edges, where $r$ represents the proportion of deleted edges in the graph. In this way, $F_{acc}^+(r)$ for positive perturbation and $F_{acc}^-(r)$ for negative perturbation can be computed as

$$F_{acc}^+(r) = \frac{1}{|T|} \sum_{i=1}^{T} \mathbb{I}(\hat{y}_i^{1-M_r^+} = y_i)),$$

$$F_{acc}^-(r) = \frac{1}{|T|} \sum_{i=1}^{T} \mathbb{I}(\hat{y}_i^{1-M_r^-} = y_i)).$$

From the above definitions we can see $F_{acc}^+$ and $F_{acc}^-$ are functions of $r$, our metrics $F_{slope}^+$ and $F_{slope}^-$ is the slop of these two metrics respectively. For example, to calculate $F_{slope}^+$ (it is similar for $F_{slope}^-$) practically, we take different values of $r$ ranging from 0 to 0.5 with step 0.1, obtaining different values of $F_{acc}^-$ for each $r$. Then we fit a linear regression to these values and get the slope, which will be $F_{slope}^+$. Clearly, $F_{slope}^+$ is a positive indicator, with higher values indicating that the model is sensitive to important features. On the other hand, $F_{slope}^-$ is a negative indicator, showing the model's insensitivity to less important features.

Table 1: Results of g-JSD and g-TVD of applying different methods to various base models before and after perturbation. ↑ means a higher value under this metric indicates better results, and ↓ means the opposite. The best performance is **bolded**. The same symbols are used in the following tables by default. ([1] Due to the substantial computational cost of calculating Laplacian positional encoding (214GiB), GT is unable to run on the ogbn-arXiv dataset.)

| Model | Method | Amazon-Photo | | Amazon-CS | | Coauthor-CS | | Coauthor-Physics | | ogbn-arXiv | |
|---|---|---|---|---|---|---|---|---|---|---|---|
| | | g-JSD↓ | g-TVD↓ | g-JSD | g-TVD | g-JSD | g-TVD | g-JSD | g-TVD | g-JSD | g-TVD |
| **GAT** | Vanilla | 6.67±1.9E-7 | 7.41±2.2 | 3.03±4.3E-7 | 24.4±15.8 | 2.29±0.5E-6 | 5.07±1.5 | 5.31±1.8E-7 | 1.77±0.4 | 4.27E-8 | 56.97 |
| | LN | 1.45±0.1E-7 | **0.06**±0.1 | 6.98±0.1E-8 | **0.03**±0.0 | **5.10**±0.1**E-7** | **0.03**±0.0 | 1.16±0.1E-7 | **0.10**±0.1 | 4.57E-7 | **0.09** |
| | AT | 1.46±0.8E-7 | 0.78±0.6 | **6.87**±0.2**E-8** | 0.34±0.1 | 1.34±1.8E-6 | 1.58±1.2 | 8.18±4.2E-7 | 1.16±0.6 | **3.89E-8** | 1.13 |
| | **FGAI** | **1.38**±0.0**E-7** | 0.46±0.1 | 7.28±0.1E-8 | 0.68±0.1 | 5.17±0.0E-7 | 1.24±0.0 | **1.10**±0.0**E-7** | 0.18±0.0 | **3.89E-8** | 3.28 |
| **GATv2** | Vanilla | 8.77±5.7E-7 | 4.63±1.9 | 1.40±0.6E-7 | 16.8±13 | 1.16±0.2E-6 | 5.07±1.5 | 3.05±0.6E-7 | 1.70±0.5 | 5.80E-8 | 3.92 |
| | LN | 2.56±2.6E-7 | 2.58±5.7 | 7.05±0.2E-8 | **0.07**±0.1 | **5.11**±0.0**E-7** | **0.03**±0.0 | **1.11**±0.0**E-7** | **0.01**±0.0 | 2.38E-7 | **0.01** |
| | AT | 1.43±0.0E-7 | 0.42±0.1 | **6.94**±0.0**E-8** | 0.23±0.1 | 6.74±1.1E-7 | 0.79±1.0 | 1.73±0.5E-7 | 0.55±0.5 | 4.92E-8 | 2.09 |
| | **FGAI** | **1.37**±0.0**E-7** | **0.28**±0.0 | 7.03±0.1E-8 | 0.47±0.1 | 5.32±0.0E-7 | 0.84±0.0 | 1.19±0.0E-7 | 0.20±0.0 | **3.96E-8** | 1.08 |
| **GT**[1] | Vanilla | 2.23±0.2E-7 | 0.45±0.3 | 9.43±1.5E-8 | 3.59±0.4 | 8.78±1.2E-7 | 12.8±5.4 | 1.13±0.1E-7 | 3.01±0.8 | OOM | OOM |
| | LN | 1.66±0.1E-7 | 1.47±0.3 | 3.50±1.7E-7 | 4.17±3.1 | **2.37**±0.0**E-7** | **0.17**±0.1 | 6.57±1.3E-8 | 0.09±0.1 | OOM | OOM |
| | AT | 2.13±0.3E-7 | 3.39±2.5 | 9.42±3.6E-7 | 0.32±0.2 | 7.73±0.1E-7 | 0.24±0.1 | **6.16**±1.4**E-8** | **0.05**±0.1 | OOM | OOM |
| | **FGAI** | **1.61**±0.0**E-7** | **0.06**±0.0 | **6.70**±0.2**E-8** | **0.28**±0.1 | 7.97±1.2E-7 | 0.30±0.1 | 8.91±0.1E-8 | 0.08±0.1 | OOM | OOM |

Table 2: Results of micro-averaged F1 scores of applying different methods to various base models before and after perturbation.

| Model | Method | Amazon-Photo | | Amazon-CS | | Coauthor-CS | | Coauthor-Physics | | ogbn-arXiv | |
|---|---|---|---|---|---|---|---|---|---|---|---|
| | | $F$↑ | $\tilde{F}1$↑ | $F$ | $\tilde{F}1$ | $F$ | $\tilde{F}1$ | $F$ | $\tilde{F}1$ | $F$ | $\tilde{F}1$ |
| **GAT** | Vanilla | 0.8206±0.07 | 0.2363±0.02 | 0.7734±0.05 | 0.3655±0.07 | 0.9006±0.00 | 0.6342±0.01 | 0.9481±0.00 | 0.7312±0.01 | 0.6813 | 0.0720 |
| | LN | 0.4429±0.01 | 0.4385±0.02 | 0.4840±0.08 | 0.4810±0.08 | 0.7268±0.01 | 0.7248±0.01 | 0.8967±0.01 | 0.8885±0.02 | 0.0729 | 0.0733 |
| | AT | 0.8451±0.05 | 0.8229±0.05 | 0.7584±0.09 | 0.7564±0.09 | **0.9069**±0.00 | 0.8398±0.10 | 0.9489±0.00 | 0.7891±0.09 | **0.6830** | 0.6742 |
| | **FGAI** | **0.8665**±0.01 | **0.8637**±0.01 | **0.8246**±0.01 | **0.8247**±0.01 | 0.8985±0.00 | **0.8905**±0.00 | **0.9493**±0.00 | **0.9485**±0.00 | 0.6760 | **0.6760** |
| **GATv2** | Vanilla | 0.8007±0.02 | 0.3655±0.16 | 0.7435±0.03 | 0.3799±0.12 | 0.9005±0.00 | 0.6558±0.02 | 0.9407±0.00 | 0.7231±0.02 | 0.6127 | 0.5523 |
| | LN | 0.4111±0.05 | 0.4093±0.05 | 0.5173±0.03 | 0.5154±0.03 | 0.7429±0.01 | 0.7406±0.01 | 0.7548±0.02 | 0.7529±0.02 | 0.0718 | 0.0718 |
| | AT | **0.8835**±0.01 | **0.8751**±0.02 | 0.7401±0.06 | 0.7419±0.06 | **0.9044**±0.00 | 0.8513±0.11 | 0.9449±0.01 | 0.8592±0.11 | 0.6085 | 0.5844 |
| | **FGAI** | 0.8160±0.02 | 0.8252±0.01 | **0.7778**±0.00 | **0.7744**±0.00 | 0.9036±0.00 | **0.8978**±0.00 | **0.9485**±0.00 | **0.9474**±0.00 | **0.6152** | **0.6122** |
| **GT** | Vanilla | 0.8501±0.03 | 0.8120±0.03 | 0.8611±0.01 | 0.8300±0.01 | 0.9042±0.01 | 0.8708±0.01 | 0.9383±0.01 | 0.9218±0.01 | OOM | OOM |
| | LN | 0.5947±0.10 | 0.4028±0.09 | 0.5492±0.12 | 0.5270±0.12 | 0.8642±0.02 | 0.8657±0.02 | 0.7602±0.15 | 0.7597±0.15 | OOM | OOM |
| | AT | 0.4988±0.11 | 0.5369±0.07 | 0.3744±0.00 | 0.3744±0.00 | 0.6403±0.18 | 0.6408±0.17 | 0.8831±0.14 | 0.8830±0.14 | OOM | OOM |
| | **FGAI** | **0.8842**±0.01 | **0.8737**±0.01 | **0.8778**±0.00 | **0.8725**±0.00 | **0.9160**±0.00 | **0.9090**±0.00 | **0.9419**±0.02 | **0.9380**±0.02 | OOM | OOM |

## 5.2 Stability Evaluation

We first conduct a comprehensive assessment of the stability against perturbations in interpretability and the output performance of FGAI in comparison to other baseline methods. In this setting, we primarily utilize g-JSD to measure attention vector stability (stability of interpretability) and g-TVD to evaluate the stability of output distributions (stability of prediction). To assess the stability of node interpretation, we initiate the testing process by randomly selecting a small number of specific nodes within the graph and introducing perturbations by adding edges and neighboring nodes to these selected nodes. We then calculate the g-JSD of the attention vectors of these two cases. Similarly, we can also evaluate the g-TVD of the two output distributions.

The results are presented in Table 1 and Table 2. We can observe that FGAI achieves the best F1 score on almost all base models and datasets, both before and after perturbations. This indicates the stability of its performance. Moreover, FGAI exhibits smaller g-JSD values compared to the vanilla model, signifying enhanced attention stability, while a similar trend is observed in the g-TVD metrics, demonstrating the stability of predictions. These findings collectively emphasize the effectiveness of FGAI in bolstering both attention and prediction stability, thereby positioning it as a reliable and robust approach for graph interpretation. However, while LN and AT exhibit lower g-JSD and g-TVD values than FGAI on some datasets, their F1 scores are very low. This suggests that they do not guarantee the stability of the performance of the base model.

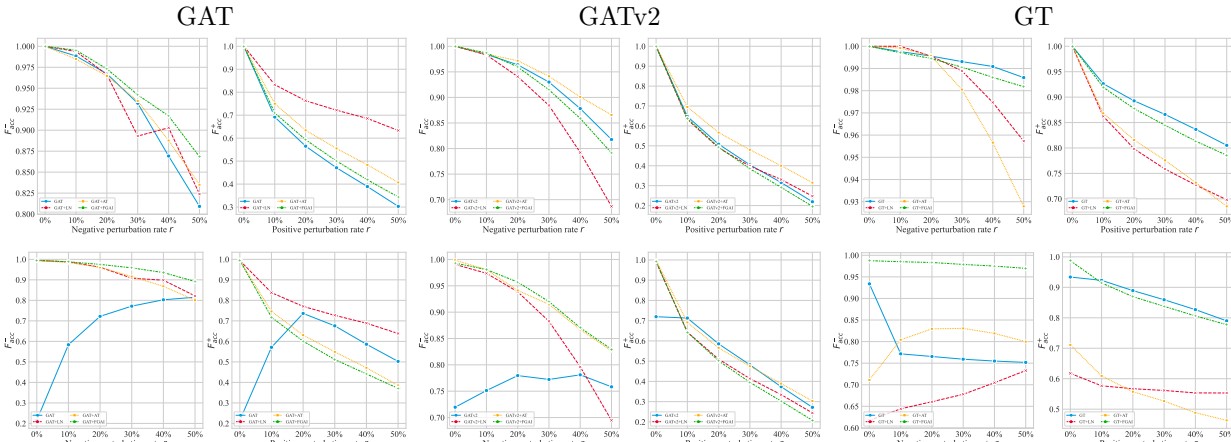

Figure 2: Results on Amazon-Photo dataset under positive and negative perturbations for LN, AT, and FGAI to GAT, GATv2, and GT, both on the clean graph (the upper figure) and the graph attacked by an injection attack (the lower figure).

## 5.3 Interpretability Evaluation

We calculate the proportion (dependent variable, i.e., $F_{acc}^+$ and $F_{acc}^-$ in the figure) of nodes that the model originally predicts correctly and still predicts correctly after removing edges based on attention weights according to the specified proportion (independent variable, i.e., $r$ in the figure). Figure 2 presents our experimental results regarding positive and negative perturbations. Due to space limitations, more results can be found in Appendix E (Figure 5, 6, 7 and Table 8). On the clean graph, all methods across all models exhibit interpretability: the proportion of nodes predicted correctly in the face of negative perturbation decreases much less than when facing positive perturbation. However, on the graph after an slight injection attack, we can observe that vanilla GAT and GATv2 show a positive slope when facing negative perturbation, indicating a complete loss of interpretability in the presence of perturbations. The same situation occurs with GT+LN and GT+AT, and since LN and AT already exhibit a significant decrease in prediction accuracy, they are not faithful interpretations.

Only FGAI, across all base models, demonstrates interpretability on both clean and attacked graphs: its performance decreases slowly when facing negative perturbation and rapidly when facing positive perturbation. This indicates that even on perturbed graphs, FGAI can ensure stable attention distributions from the base model, providing a faithful explanation. Due to space constraints, additional analyses on the experimental results can be found in Appendix E.

## 5.4 Visualization Results

Our visualization results come in two forms: (1) We showcase the attention values of a selected subset of nodes and edges from the graph data before and after perturbation, as depicted in Figure 3. (2) We highlight the top-$k$ most important neighboring nodes and edges connected to a specific node before and after perturbation, as presented in Figure 3. Due to space limitations, some of the visualization results are presented in Appendix F (Figure 8). In Figure 3, we observe that the attention values of GAT significantly decrease after perturbation, compromising the topological importance structure of the graph and thus losing their interpretability for the graph. In contrast, FGAI maintains relatively consistent attention values before and after perturbation, demonstrating its superior performance in resisting perturbations while preserving interpretable stability. Likewise, Figure 3 reinforces our model's stability in retaining the most crucial characteristics of neighboring node information, reinforcing the faithfulness of our approach.

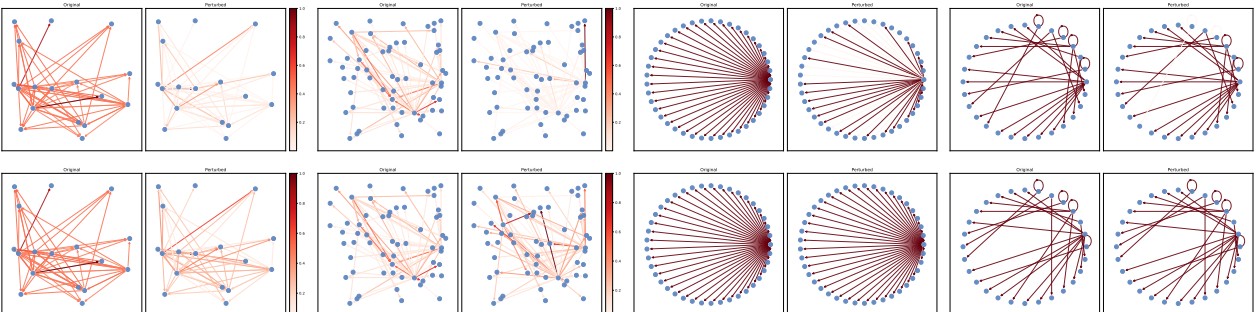

Figure 3: (a) Left Four Column: Visualizations of the attention results for GAT and FGAI, showcasing a subset of nodes and edges from the graph data before and after perturbation. The color of the edges corresponds to their respective magnitude of values. (b) Right Four Column: Visualizations of the attention results for GAT and FGAI before and after perturbation. The red color connects the nodes that appeared in top-$k$ nodes.

### 5.5 Ablation Study and Computation Cost

We also conducted experiments on the ablation study in Table 9 as well as time and storage complexities comparison in Table 10, see Appendix H and I for details. These findings reinforce the central roles played by the top-$k$ loss and TVD loss in improving the faithfulness of the model. For computational cost, we observe that, compared to these methods, FGAI is a more efficient approach. Our method incurs relatively minimal additional overhead compared to the vanilla model, offering an efficient and cost-saving solution for the graph defense community.

## 6 Conclusions

While some studies (Shin et al., 2024) argue that attention mechanisms inherently provide interpretability, others (Fan et al., 2021; Panagiotaki et al., 2023) highlight their limitations, such as instability or misalignment with model behavior. This debate remains unresolved, especially in GNNs, where graph structure adds complexity. Our work does not aim to settle this broader question but instead focuses on a practical goal: if attention is used for explanations in Attention-based GNNs (as commonly done), *how can we improve its faithfulness?* In this study, we investigated the faithfulness issues in Attention-based GNNs and proposed a rigorous definition for FGAI. FGAI is characterized by four key properties emphasizing stability and sensitivity, making it a more reliable tool for graph interpretation. To assess our approach rigorously, we introduced two novel evaluation metrics for graph interpretations. Results show that FGAI excels in preserving interpretability while enhancing stability, outperforming other methods under perturbations and adaptive attacks.

### Acknowledgments

Di Wang and Lijie Hu are supported in part by the funding BAS/1/1689-01-01, URF/1/4663-01-01, REI/1/5232-01-01, REI/1/5332-01-01, and URF/1/5508-01-01 from KAUST, and funding from KAUST - Center of Excellence for Generative AI, under award number 5940. Wanyu Lin is supported in part by Hong Kong (HK) Research Grant Council (RGC) General Research Fund under Grant PolyU 15208222, NSFC Young Scientist Fund under Grant PolyU A0040473.

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

## A  Optimization

In the definition section, we presented a rigorous definition of faithful node-level graph interpretation. To find such an FGAI, we propose to formulate a min-max optimization problem that involves the four conditions in Definition 3.2. Specifically, the formulated optimization problem takes the third condition (closeness of prediction) as the objective and subjects it to the other three conditions. Thus, we can get a rough optimization problem according to the definition. Specifically, we first have

$$\min_{\tilde{w}} \mathbb{E}_h D(y(h, \tilde{w}), y(h, w)). \tag{4}$$

Equation (4) is the basic optimization goal. That is, we want to get a vector that has a similar output prediction with vanilla GAT for all input nodes $h$. If there is no further constraint, then we can see the minimizer of (4) is just the vanilla GAT $w$. We then consider constraints for this objective function:

$$\forall h \text{ s.t. } \max_{||\delta|| \leq R} D(y(h, \tilde{w}), y(h, \tilde{w} + \delta)) \leq \alpha, \tag{5}$$

$$V_{k_1}(\tilde{w}(h), w(h)) \geq \beta, \tag{6}$$

$$\max_{||\rho|| \leq R} V_{k_2}(\tilde{w}(h), \tilde{w}(h) + \rho) \geq \beta. \tag{7}$$

Equation (5) is the constraint of stability, Equation (6) corresponds to the condition of similarity of explanation, and Equation (7) links to the stability of explanation. Combining equations (4)-(7) and using regularization to deal with constraints, we can get the following objective function.

$$\min_{\tilde{w}} \mathbb{E}_h[D(y(h, \tilde{w}), y(h, w)) + \lambda_1 (\beta - V_{k_1}(\tilde{w}(h), w(h)))$$
$$+ \lambda_2 (\max_{||\delta|| \leq R} D(y(h, \tilde{w}), y(h, \tilde{w} + \delta)) - \alpha) + \lambda_3 \max_{||\rho|| \leq R} (\beta - V_{k_2}(\tilde{w}(h), \tilde{w}(h) + \rho))], \tag{8}$$

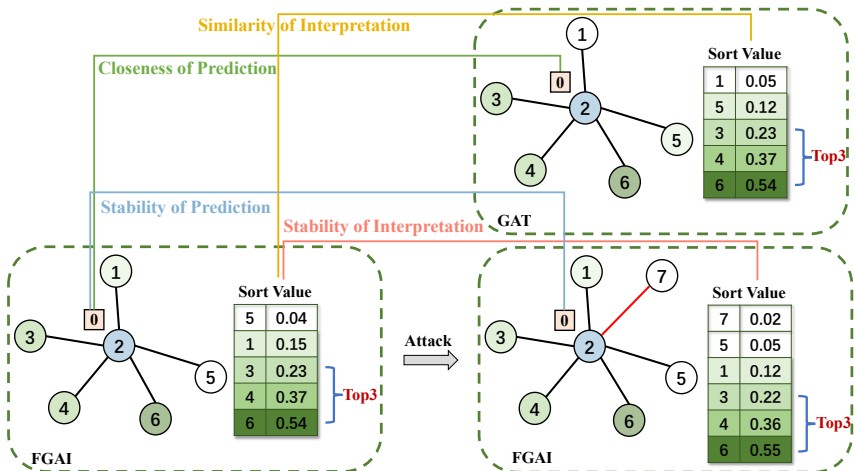

Figure 4: The illustration of our definition.

where $\lambda_1 > 0$, $\lambda_2 > 0$ and $\lambda_3 > 0$ are hyperparameters.

From now on, we convert the problem of finding a vector that satisfies the four conditions in Definition 3.2 to a min-max stochastic optimization problem, where the overall objective is based on the closeness of prediction condition with constraints on stability and top-$k$ overlap.

Next, we consider how to solve the above min-max optimization problem. In general, we can use the stochastic gradient descent-based methods to get the solution of outer minimization and use PSGD (Projected Stochastic Gradient Descent) to solve the inner maximization. However, the main difficulty is that the top-$k$ overlap function $V_{k_1}(\tilde{w}(h), w(h))$ and $V_{k_2}(\tilde{w}(h), \tilde{w}(h) + \rho)$ is non-differentiable, which impede us from using gradient descent. Thus, we need to consider a surrogate loss of $-V_k(\cdot)$. Below, we provide details.

**Projected gradient descent to find the perturbation $\delta$.** Motivated by Madry et al. (2018), we can interpret the perturbation as the attack to $\tilde{w}$ via maximizing $\delta$. Then, $\delta$ can be updated by the following procedure in the $p$-th iteration.

$$\delta_p = \delta^*_{p-1} + \gamma_p \frac{1}{|B_p|} \sum_{h \in B_p} \nabla D(y(h, \tilde{w}), y(h, \tilde{w} + \delta^*_{p-1}));$$

$$\delta^*_p = \underset{||\delta|| \leq R}{\arg\min} ||\delta - \delta_p||,$$

where $\gamma_p$ is a parameter of step size for PGD, $B_p$ is a batch and $|B_p|$ is the batch size. Using this method, we can derive the optimal $\delta^*$ in the $t$-th iteration of outer minimization for the inner optimization. Specifically, we find a $\delta$ as the maximum tolerant of perturbation w.r.t $\tilde{\boldsymbol{w}}$ in the $t$-th iteration of outer SGD.

**Top-$k$ overlap surrogate loss.** Now, we seek to design a surrogate loss $\mathcal{L}_k(\cdot)$ for $-V_k(\cdot)$ which can be used in training. We takes the $\mathcal{L}_k(\tilde{w})$ for $-V_k(\tilde{w}, w)$ as an example. To achieve this goal, one possible naive surrogate objective might be some distance (such as $\ell_1$-norm) between $\tilde{w}$ and $w$, e.g., $L(\tilde{w}) = ||\tilde{w} - w||_1$. Such a surrogate objective seems like it could ensure the top-$k$ overlap when we obtain the optimal or near-optimal solution (i.e., $w = \arg\min L(\tilde{w})$ and $w \in \arg\min -V_{k_1}(\tilde{w}, w)$). However, it lacks consideration of the top-$k$ information, which makes it a loose surrogate loss. Since we only need to ensure high top-$k$ indices overlaps between $\tilde{w}$ and $w$, one improved method is minimizing the distance between $\tilde{w}$ and $w$ constrained on the top-$k$ entries only instead of the whole vectors, i.e., $||w_{S^k_w} - \tilde{w}_{S^k_w}||_1$, where $w_{S^k_w}, \tilde{w}_{S^k_w} \in \mathbb{R}^k$ is the vector $w$ and $\tilde{w}$ constrained on the indices set $S^k_w$ respectively and $S^k_w$ is the top-$k$ indices set of $w$. Since there are two top-$k$ indices sets, one is for $\tilde{w}$, and the other one is for $w$, we need to use both of them to involve the top-$k$ indices formation for both vectors. Thus, based on our above idea, our surrogate can be written as follows,

---

**Algorithm 1** FGAI

---

1: **Input:** Graph $h = \{h_1, h_2, \ldots, h_N\}$, attention weight $w$.
2: Initialize faithful attention layer $\tilde{w}_0$.
3: **for** $t = 1, 2, \cdots, T$ **do**
4:     Initialize $\delta_0$, $\rho_0$.
5:     **for** $p = 1, 2, \cdots, P$ **do**
6:        Update $\boldsymbol{\delta}$ using PGD.
7:        $\delta_p = \delta_{p-1} + \gamma_p \sum \nabla D(y(h, \tilde{w}_{t-1}), y(h, \tilde{w}_{t-1} + \delta_{p-1}))$.
8:        $\delta_p^* = \underset{||\delta|| \leq R}{\arg\min} ||\delta - \delta_p||$.
9:     **end for**
10:    **for** $q = 1, 2, \cdots, Q$ **do**
11:       Update $\rho$ using PGD.
12:       $\rho_q = \rho_{q-1} - \tau_q \sum \nabla \mathcal{L}_{k_2}(\tilde{w}_{t-1}, \tilde{w}_{t-1} + \rho_{q-1})$.
13:       $\rho_q^* = \underset{||\rho|| \leq R}{\arg\min} ||\rho - \rho_q||$.
14:     **end for**
15:    Update $\tilde{w}$ using Stochastic Gradient Descent.

$$\tilde{w}_t = \tilde{w}_{t-1} - \eta_t \sum [\nabla D(y(h, \tilde{w}_{t-1}), y(h, w))$$
$$- \lambda_1 \nabla \mathcal{L}_{k_1}(w, \tilde{w}_{t-1}) + \lambda_2 \nabla D(y(h, \tilde{w}_{t-1}), y(h, \tilde{w}_{t-1} + \delta_P^*)) - \lambda_3 \nabla \mathcal{L}_{k_2}(\tilde{w}_{t-1}, \tilde{w}_{t-1} + \rho_Q^*)].$$

16: **end for**
17: **Return:** $\tilde{w}^* = \tilde{w}_T$.

---

$$\mathcal{L}_k(w, \tilde{w}) = \frac{1}{2k}(||w_{S_w^k} - \tilde{w}_{S_w^k}||_1 + ||\tilde{w}_{S_{\tilde{w}}^k} - w_{S_{\tilde{w}}^k}||_1). \tag{9}$$

Note that besides the $\ell_1$-norm, we can use other norms. However, in practice, we find $\ell_1$-norm achieves the best performance. Thus, throughout the paper, we only use $\ell_1$-norm.

**Projected gradient descent to find the perturbation $\rho$.** Similarly, we can use the PGD and the surrogate loss of $\mathcal{L}_k(\cdot)$ to get the optimal $\rho^*$ in the $t$-th iteration of outer SGD.

$$\rho_q = \rho_{q-1}^* + \tau_q \frac{1}{|B_q|} \sum_{h \in B_q} \nabla \mathcal{L}_{k_2}(\tilde{w}, \tilde{w} + \rho_{q-1});$$

$$\rho_q^* = \underset{||\rho|| \leq R}{\arg\min} ||\rho - \rho_q||,$$

where $\tau_q$ is a parameter of step size for PGD, $B_q$ is a batch and $|B_q|$ is the batch size.

**Final objective function and algorithm.** Based on the above discussion, we can derive the following overall objective function

$$\min_{\tilde{w}} \mathbb{E}_x[D(y(h, \tilde{w}), y(h, w)) + \lambda_1 \mathcal{L}_{k_1}(w, \tilde{w})$$
$$+ \lambda_2 \max_{||\delta|| \leq R} D(y(h, \tilde{w}), y(h, \tilde{w} + \delta)) + \lambda_3 \max_{||\rho|| \leq R} \mathcal{L}_{k_2}(\tilde{w}, \tilde{w} + \rho)], \tag{10}$$

where $\mathcal{L}_k(\cdot)$ is defined in (9). Based on the previous idea, we propose Algorithm 1 to solve (10).

So far, we propose an efficient algorithm to solve this objective function, which meets the FGAI definition. The first term $D$ minimizes the output difference of vanilla GAT and FGAI to make it more stable. The second term top-$k$ is substituted by a surrogate loss, which is differentiable and practical to compute via

backpropagation. This term guarantees the explainable information of the attention. The third term $D$ is a min-max optimization controlled by hyperparameter $\lambda_2$ in order to find the maximum tolerant perturbation to the attention layer, which affects the final prediction. The final term $\mathcal{L}_{k_2}$ is also a min-max optimization to find the maximum tolerant perturbation to the intrinsic explanation of the attention layer. In other words, we derive a robust region using this min-max strategy. Thus, we get the final version (10) to solve our problem. Details of the algorithm are presented in Algorithm 1.

## B  Differences with adversarial robustness

While both FGAI and adversarial robustness consider perturbations or noises on graph data. There are many critical differences: (1) In adversarial robustness, the goal is only to make the prediction unchanged under perturbation on the input (the property of stability of prediction), while in FGAI we should additionally keep the prediction close to the vanilla one. (2) Not only the prediction, we should also make the interpretability stable and close to the vanilla attention. Due to these additional conditions, our method for achieving FGAI is totally different from the methods in adversarial robustness, such as certified robustness or adversarial training. See Section 4 for details. (3) The way of modeling robustness in FGAI is also totally different from adversarial robustness. In adversarial robustness, it usually models the robustness to perturbation on input data. However, in FGAI, due to the requirement on interpretability, i.e., the top-$k$ indices of the vector, we cannot adopt the same idea. Firstly, directly requiring the top-$k$ indices robust to perturbation on the input will make the optimization procedure challenging (which is a minimax optimization problem) as the top-$k$ indices function is non-differentiable, and calculating the gradient of the attention is costly. Secondly, rather than perturbation on input data, as we mentioned, the perturbation could come from multi-resources, such as a combined perturbation of edges and additional nodes. Thus, from this perspective, our stability of interpretability is more suitable for "stable attention".

Table 3: Statistics of the datasets used for the node classification task.

|  | Amazon-Photo | Amazon-CS | Coauthor-CS | Coauthor-Physics | ogbn-arXiv |
|---|---|---|---|---|---|
| #Nodes | 7,650 | 13,752 | 18,333 | 34,493 | 169,343 |
| #Edges | 119,043 | 245,778 | 81,894 | 247,962 | 1,166,243 |
| #Node Features | 745 | 767 | 6,805 | 8,415 | 128 |
| #Classes | 8 | 10 | 15 | 5 | 40 |
| #Training Nodes | 765 | 1,375 | 1,833 | 3,449 | 90,941 |
| #Validation Nodes | 765 | 1,375 | 1,833 | 3,449 | 29,799 |
| #Test Nodes | 6,120 | 11,002 | 14,667 | 27,595 | 48,603 |

## C  Additional Information for the Dataset

We demonstrate our dataset statistics in Table 3, Table 4 and Table 5. Amazon CS and Amazon Photo are divisions within the Amazon co-purchase network originally introduced in McAuley et al. (2015), where nodes represent products, edges signify common pairing of two products in customer purchases, and node features are based on bag-of-words encoded product reviews, with class labels determined by the product category. Coauthor CS and Coauthor Physics (Shchur et al., 2018) refers to co-authorship networks derived from the KDD Cup where individual authors are depicted as nodes, and an edge connects them if they have collaborated on a research paper. Node features encompass paper keywords associated with each author's publications, and labels are indicative of the primary areas of study in which each author is most active. The ogbn-arXiv dataset (Wang et al., 2020; Mikolov et al., 2013) represents the citation network between all Computer Science arXiv papers indexed by MAG. Each node is an arXiv paper, and each directed edge indicates that one paper cites another. The Cora dataset (Sen et al., 2008) collection contains 2,708 scholarly papers categorized into seven research domains, while the PubMed corpus comprises 19,717 diabetes-related publications from the PubMed database grouped into three thematic categories. The CiteSeer dataset includes 3,312 academic documents organized into six subject classifications. In structural biology, the D&D dataset (Debnath et al., 1991) features 1,178 high-resolution protein structures from the Protein Data Bank's non-redundant subset, where amino acid residues form nodes connected by edges when their spatial distance

Table 4: Statistics of the datasets used for graph classification task.

|  | D&D | MUTAG | Politifact |
|---|---|---|---|
| #Graphs | 1,178 | 188 | 314 |
| #Classes | 2 | 3 | 2 |
| #Avg. Nodes | 284.32 | 17.93 | 130.74 |
| #Avg. Edges | 715.66 | 19.79 | 129.75 |
| #Training Graphs | 706 | 112 | 188 |
| #Validation Graphs | 118 | 18 | 31 |
| #Test Graphs | 354 | 58 | 95 |

Table 5: Statistics of the datasets used for the link prediction task.

|  | Cora | Pubmed | Citeseer |
|---|---|---|---|
| #Nodes | 2,708 | 19,717 | 3,327 |
| #Edges | 9,502 | 79,784 | 8,194 |
| #Node Features | 1,433 | 500 | 3,703 |
| #Training Links | 4,488 | 37,676 | 3,870 |
| #Validation Links | 526 | 4,432 | 454 |
| #Test Links | 1,054 | 8,864 | 910 |

is below 6Å. For chemoinformatics applications, MUTAG (Dobson & Doig, 2003) provides a repository of nitroaromatic compounds annotated for mutagenicity assessment in Salmonella typhimurium. Regarding social media analysis, the Politifact (Dou et al., 2021) dataset captures Twitter propagation networks of news stories labeled as factual or deceptive based on professional fact-checking evaluations.

# D  Additional Results on Stability Evaluation

Table 6: The results of g-JSD, g-TVD, and classification accuracy before and after perturbation when applying different methods to various base models in the graph classification task.

| Model | Method | D&D | | | | MUTAG | | | | Citeseer | | | |
|---|---|---|---|---|---|---|---|---|---|---|---|---|---|
| | | g-JSD↓ | g-TVD↓ | ROC-AUC↑ | attacked↑ | g-JSD | g-TVD | ROC-AUC | attacked | g-JSD | g-TVD | ROC-AUC | attacked |
| **GAT** | Vanilla | 6.38E-07 | 1.22 | 0.7327 | 0.6258 | 2.59E-05 | 0.35 | **0.7945** | 0.6849 | 2.77E-06 | 1.30 | **0.9000** | 0.8667 |
| | **FGAI** | **2.58E-08** | **0.19** | **0.7338** | **0.6949** | **2.06E-05** | **0.26** | 0.7534 | **0.7301** | **2.38E-06** | 1.13 | 0.8967 | **0.8833** |
| **GATv2** | Vanilla | 8.07E-07 | 1.42 | 0.7528 | 0.5857 | 2.62E-05 | 0.40 | 0.7671 | 0.6164 | 2.34E-06 | 1.00 | 0.8833 | 0.8083 |
| | **FGAI** | **6.00E-08** | **0.17** | **0.7582** | **0.7404** | **1.32E-05** | **0.20** | **0.7945** | **0.7549** | **2.34E-06** | **0.88** | **0.9000** | **0.8800** |

In Table 6 and Table 7, we present a comparison of the performance of our method and the base model on the graph classification and link prediction task. As shown, our method significantly enhances the stability of the attention layer, resulting in the base model demonstrating robust stability when faced with small perturbations.

# E  Additional Results on Interpretability Evaluation

Here, we present the results of the interpretability evaluation on all datasets. Similar to before, we showcase the results in the face of positive and negative perturbations when applying LN, AT, and FGAI to GAT, GATv2, and GT, both on the clean graph (the upper figure) and the graph attacked by an injection attack (the lower figure) on various datasets. We calculate the proportion (dependent variable, i.e., $F_{acc}^+$ and $F_{acc}^-$ in the figure) of nodes that the model originally predicts correctly and still predicts correctly after removing important or unimportant edges based on attention weights (i.e., positive perturbation or negative perturbation) according to the specified proportion (independent variable, i.e., $r$ in the figure).

From the results shown in Figure 5, 6 and 7, supplemented by Table 8, we can more comprehensively examine the interpretability of the models after applying different methods to different models and draw the following conclusions:

Table 7: The results of g-JSD, g-TVD, and the ROC-AUC score before and after perturbation when applying different methods to various base models in the link prediction task.

| Model | Method | Cora | | | | Pubmed | | | | Citeseer | | | |
|---|---|---|---|---|---|---|---|---|---|---|---|---|---|
| | | g-JSD↓ | g-TVD↓ | ROC-AUC↑ | attacked↑ | g-JSD | g-TVD | ROC-AUC | attacked | g-JSD | g-TVD | ROC-AUC | attacked |
| GAT | Vanilla | 2.23E-05 | 0.0074 | **0.8502** | 0.7855 | 4.08E-06 | 0.0063 | **0.8268** | 0.7203 | 4.69E-05 | 0.0308 | **0.8424** | 0.7115 |
| | FGAI | **1.79E-05** | **0.0065** | 0.8362 | **0.8107** | **3.88E-06** | **0.0008** | 0.8248 | **0.8153** | **3.17E-05** | **0.0060** | 0.8346 | **0.7873** |
| GATv2 | Vanilla | 1.84E-05 | 0.0046 | 0.8678 | 0.8320 | 3.87E-06 | 0.0006 | 0.8377 | 0.8328 | 4.57E-05 | 0.0201 | **0.8300** | 0.7535 |
| | FGAI | **1.80E-05** | **0.0027** | **0.8741** | **0.8589** | **3.81E-06** | **0.0004** | **0.8414** | **0.8345** | **3.17E-05** | **0.0099** | 0.8181 | **0.7870** |

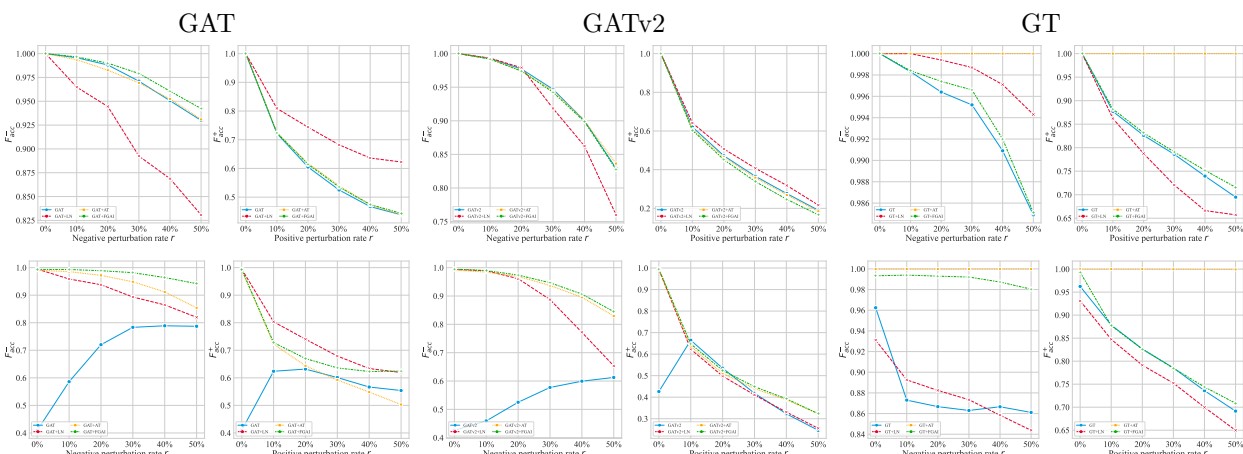

Figure 5: Additional results of interpretability evaluation on Amazon-CS.

- Whether it is GAT, GATv2, or GT, they all exhibit some interpretability on clean graphs. When facing negative perturbation, GT performs the best—its performance decreases the least with an increase in perturbation rate $r$, maintaining $F_{acc}^-$ above 0.95 across different datasets. GAT and GATv2, on the other hand, show a more pronounced decrease, indicating a weaker insensitivity to unimportant edges. When facing positive perturbation, the $F_{acc}^+$ of GAT and GATv2 sharply declines with the perturbation rate $r$, surpassing GT, demonstrating sensitivity to important edges (edges with larger attention values).

- On a clean graph, the performance of different methods (LN, AT, and FGAI) is similar. Moreover, the performance of FGAI is almost identical to the vanilla model, confirming that FGAI ensures similarity of interpretability and closeness of prediction with the vanilla model.

- On the attacked graph, the interpretability of all vanilla Attention-based GNNs is compromised, with abnormal phenomena such as positive slopes (improved performance as edges decrease). LN and AT also fail to maintain the interpretability of the base model. Only FGAI still ensures the stability of interpretability and stability of prediction. Moreover, at this point, the $F_{slope}^+$ of FGAI is consistently greater than that of the base model across all datasets, while $F_{slope}^-$ is consistently smaller than the base model, demonstrating outstanding interpretability.

## F  Additional Visualization

Here, we present additional visualization results in Figure 8.

## G  Implementation Details

Regarding the architecture of the vanilla model, for the ogbn-arxiv dataset, we configure GAT, GATv2 and GT with three layers and 8 attention heads. The hidden layer dimension is set to 128. On other datasets,

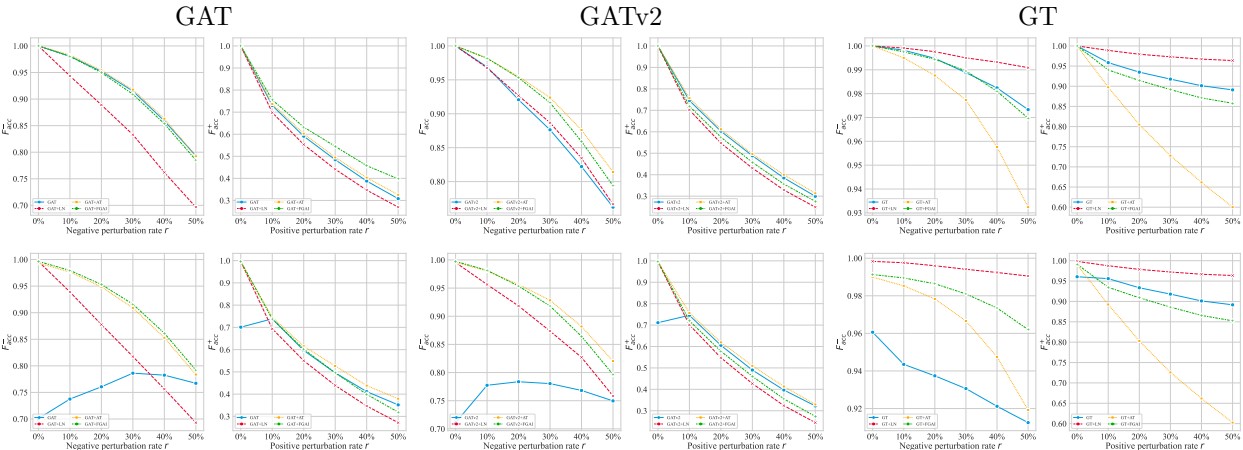

Figure 6: Additional results of interpretability evaluation on Coauthor-CS.

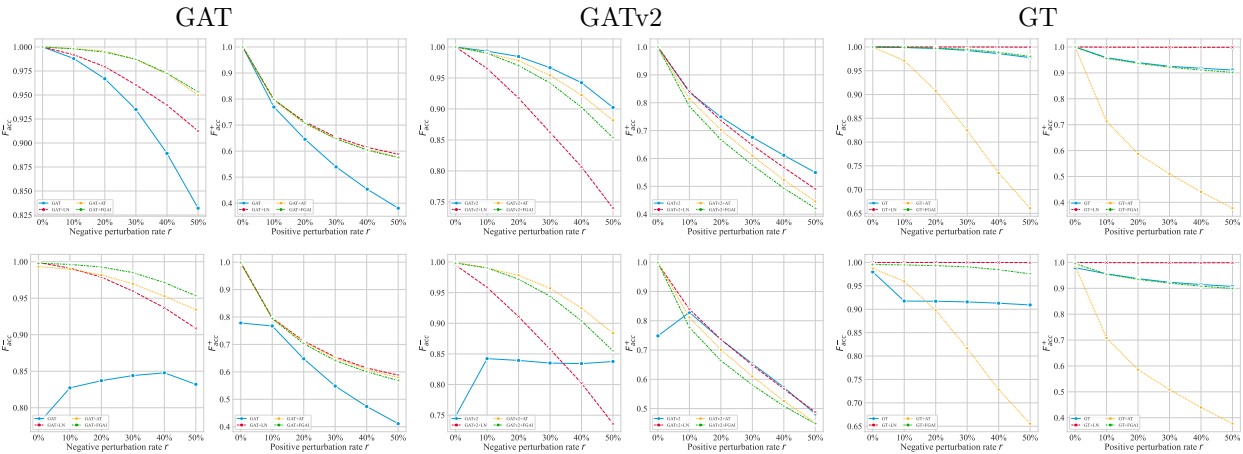

Figure 7: Additional results of interpretability evaluation on Coauthor-Physics.

we use one layer and an 8-dimensional hidden layer with 8 attention heads for GAT and GATv2 and an 128-dimensional hidden layer with 8 attention heads for GT. A consistent feature dropout rate and an attention dropout rate of 0.6 are applied to all datasets. We use Adam as the optimizer. The learning rate is fixed at 0.01, and an $L_2$ regularization is set to 0.0005. During the training of FGAI, we maintained the same parameters as the vanilla model. For the Coauthor-CS and Coauthor-Physics datasets, we set $\lambda_1$, $\lambda_2$, and $\lambda_3$ to 1, 1, and 1, respectively, with $K$ values of 100,000 and 200,000, respectively. For the Amazon-CS and Amazon-Photo datasets, $\lambda_1$, $\lambda_2$, and $\lambda_3$ were set to 0.8, 1, and 1, and $K$ was set to 100,000. On the ogbn-arXiv dataset, we used $\lambda_1$, $\lambda_2$, and $\lambda_3$ values of 1, 10, and 5, respectively, with $K$ set to 600,000. Note that for the hyperparameter $K$, we can observe significant differences in its values across different datasets. The reason is that according to the definition of FGAI in Definition 3.2, we need to find an appropriate $K$ value to ensure that the FGAI attention vector can retain the most critical features of the vanilla attention vector while relaxing its constraints on less important features. This preserves the interpretability of attention under different forms of perturbations and randomness. Therefore, we set $K$ to approximately 50% of the number of edges in the dataset (the total length of the attention vector). In Figure 9, we show the line chart of different components of the loss of FGAI on various datasets as the epoch increases. It can be seen that the stability of explanation loss corresponding to $\lambda_2$ and the similarity of explanation loss corresponding to $\lambda_3$ remain at a relatively low value.

Table 8: The results of our proposed metrics, $F_{slope}^{+}$ and $F_{slope}^{-}$, for evaluating the interpretability of different methods applied to different models across all datasets. Note that $F_{slope}^{+}$ and $F_{slope}^{-}$ are generally negative values; the numbers in the table are displayed as absolute values. Values that were originally positive are marked with an underline.

| Model | Method | Before Attack | | | | | | | | After Attack | | | | | | | |
| | | Amazon-Photo | | Amazon-CS | | Coauthor-CS | | Coauthor-Physics | | Amazon-Photo | | Amazon-CS | | Coauthor-CS | | Coauthor-Physics | |
| | | $F-\downarrow$ | $F+\uparrow$ | $F-$ | $F+$ | $F-$ | $F+$ | $F-$ | $F+$ | $F-$ | $F+$ | $F-$ | $F+$ | $F-$ | $F+$ | $F-$ | $F+$ |
| **GAT** | Vanilla | 0.3848 | 1.2821 | 0.1437 | 1.0437 | 0.4101 | 1.3123 | 0.3337 | 1.1857 | 1.0594 | 0.4061 | 0.7312 | 0.1485 | 0.1406 | 0.8078 | 0.0967 | 0.8045 |
| | LN | 0.3504 | 0.6613 | 0.3389 | 0.7050 | 0.6050 | 1.3766 | 0.1754 | 0.7609 | 0.3385 | 0.6481 | 0.3431 | 0.6991 | 0.6071 | 1.3644 | 0.1811 | 0.7594 |
| | AT | 0.3275 | 1.0979 | 0.1376 | 1.0409 | 0.4098 | 1.2803 | 0.0963 | 0.7821 | 0.3881 | 1.1285 | 0.2648 | 0.8607 | 0.4168 | 1.1650 | 0.1196 | 0.7708 |
| | **FGAI** | 0.2640 | 1.2135 | 0.1162 | 1.0315 | 0.4272 | 1.1406 | 0.0906 | 0.7917 | 0.1976 | 1.1548 | 0.1007 | 0.6269 | 0.4041 | 1.2884 | 0.0871 | 0.7972 |
| **GATv2** | Vanilla | 0.3609 | 1.4293 | 0.3320 | 1.4822 | 0.4782 | 1.3420 | 0.1882 | 0.8579 | 0.0787 | 0.9599 | 0.4016 | 0.5834 | 0.0463 | 0.8871 | 0.1193 | 0.6239 |
| | LN | 0.6270 | 1.3605 | 0.4724 | 1.4244 | 0.4589 | 1.4260 | 0.5235 | 0.9873 | 0.5908 | 1.3582 | 0.6933 | 1.3302 | 0.4623 | 1.4303 | 0.5190 | 0.9825 |
| | AT | 0.2723 | 1.2560 | 0.3207 | 1.4861 | 0.3650 | 1.3192 | 0.2338 | 1.0677 | 0.3505 | 1.2787 | 0.3166 | 1.1877 | 0.3432 | 1.2775 | 0.2272 | 1.0524 |
| | **FGAI** | 0.4206 | 1.4765 | 0.3349 | 1.5294 | 0.4102 | 1.3793 | 0.2920 | 1.1040 | 0.3389 | 1.4440 | 0.2935 | 1.1995 | 0.3975 | 1.3801 | 0.2869 | 1.0392 |
| **GT** | Vanilla | 0.0266 | 0.3630 | 0.0283 | 0.5655 | 0.0534 | 0.2098 | 0.0445 | 0.1663 | 0.2761 | 0.2964 | 0.1513 | 0.5209 | 0.0898 | 0.1499 | 0.1052 | 0.1431 |
| | LN | 0.0846 | 0.5585 | 0.0108 | 0.6764 | 0.0188 | 0.0720 | 0.0007 | 0.0023 | 0.2216 | 0.1136 | 0.1566 | 0.5396 | 0.0161 | 0.0691 | 0.0007 | 0.0023 |
| | AT | 0.1438 | 0.5783 | 0.0000 | 0.0000 | 0.1316 | 0.7955 | 0.7109 | 1.1448 | 0.1398 | 0.4663 | 0.0000 | 0.0012 | 0.1367 | 0.7733 | 0.6952 | 1.1237 |
| | **FGAI** | 0.0366 | 0.4064 | 0.0270 | 0.5293 | 0.0588 | 0.2691 | 0.0382 | 0.1828 | 0.0354 | 0.4017 | 0.0244 | 0.5328 | 0.0571 | 0.2640 | 0.0379 | 0.1815 |

Figure 8: (a) Left Four Column: Additional visualizations of the attention results for both the vanilla model (GAT) and FGAI on Coauthor-CS and Coauthor-Physics dataset, showcasing a subset of nodes and edges from the graph data before and after perturbation. We have selected a representative portion of the graph for display due to the substantial size of the dataset. The color of the edges corresponds to their respective magnitude of values. (b) Right Four Column: Additional visualizations of the attention results for both the vanilla model (GAT) and FGAI on two datasets, showcasing top-$k$ important neighboring nodes and edges from a specific node before and after perturbation. The red color connects the nodes both appeared in top-$k$ nodes before and after perturbation.

For the perturbation method, we first randomly generate $n$ nodes and connect them with $e$ random edges, and then employ the Projected Gradient Descent algorithm to create features for these new nodes, maximizing the perturbation for the graph. In our experiments, we set $n=20$, $e=20$, and the perturbation level on features to be 0.1. Additionally, we utilize Total Variation Distance loss as the loss term $D$ in Algorithm 1 and select the $\ell_1$-norm as the perturbation radius norm.

For all experimental data, we conduct five runs using different random seeds and reported both the average results and standard deviations.

# H Ablation Study

We conduct ablation experiments on the loss function to demonstrate that each component of the loss plays an indispensable role in bolstering the efficacy of FGAI. As illustrated in Algorithm 1, we systematically remove one or several components of the loss function by setting the corresponding $\lambda_i$ values to 0. Subsequently, we evaluate the performance of FGAI with the remaining components. From the results presented in Table 9,

Table 9: Ablation study of the proposed method. We evaluate the effectiveness of three loss functions in objective function on Amazon-Photo dataset.

| Model | Ablation Setting | | | Metrics | | | |
|-------|:---:|:---:|:---:|:---:|:---:|:---:|:---:|
| | $\mathcal{L}_1$ | $\mathcal{L}_2$ | $\mathcal{L}_3$ | g-JSD↓ | g-TVD↓ | $F1$↑ | $\tilde{F}1$↑ |
| | ✓ | ✓ | ✓ | 1.3808E-7 | **0.4576** | 0.8665 | 0.8637 |
| | | ✓ | ✓ | 1.3966E-7 | 0.4587 | **0.8698** | **0.8662** |
| | ✓ | | ✓ | **1.3689E-7** | 0.7937 | 0.8222 | 0.8180 |
| GAT | ✓ | ✓ | | 1.3719E-7 | 0.4676 | 0.8616 | 0.8590 |
| | | | ✓ | 1.3700E-7 | 0.7991 | 0.8273 | 0.8225 |
| | | ✓ | | 1.3880E-7 | 0.5028 | 0.8616 | 0.8587 |
| | ✓ | | | 1.3717E-7 | 0.8529 | 0.8209 | 0.8144 |
| | ✓ | ✓ | ✓ | 1.3708E-7 | **0.2804** | **0.8160** | **0.8252** |
| | | ✓ | ✓ | 1.3710E-7 | 0.3086 | 0.8111 | 0.8216 |
| | ✓ | | ✓ | **1.3680E-7** | 0.5471 | 0.8029 | 0.8183 |
| GATv2 | ✓ | ✓ | | 1.3711E-7 | 0.3390 | 0.8119 | 0.8081 |
| | | | ✓ | 1.3682E-7 | 0.5735 | 0.7493 | 0.7539 |
| | | ✓ | | 1.3714E-7 | 0.2941 | 0.8091 | 0.8206 |
| | ✓ | | | 1.3700E-7 | 0.5415 | 0.7549 | 0.7672 |
| | ✓ | ✓ | ✓ | **1.0210E-6** | 0.4295 | **0.8482** | **0.8319** |
| | | ✓ | ✓ | 2.8307E-5 | 0.5964 | 0.7377 | 0.7059 |
| | ✓ | | ✓ | 1.0275E-6 | 1.0221 | 0.8351 | 0.8090 |
| GT | ✓ | ✓ | | 1.0533E-6 | 0.3598 | 0.7779 | 0.7595 |
| | | | ✓ | 1.0281E-6 | 0.9455 | 0.8312 | 0.8087 |
| | | ✓ | | 2.9748E-5 | 0.7176 | 0.8410 | 0.8096 |
| | ✓ | | | 1.0275E-6 | 1.0120 | 0.8279 | 0.8044 |

Table 10: Time overhead and GPU memory usage for various methods on different datasets, all values are estimated, conducted on an NVIDIA V100 device with 200 epochs. OOM (Out of Memory) indicates exceeding the GPU memory limit, rendering the program unable to run.

| Method | Amazon-Photo | | Coauthor-Physics | | ogbn-arXiv | |
|--------|:---:|:---:|:---:|:---:|:---:|:---:|
| | Time | GPU | Time | GPU | Time | GPU |
| GAT | 30s | 1000MiB | 1min | 4000MiB | 2min | 4000MiB |
| GAT+LN | 30s | 1100MiB | 1min | 4000MiB | 2min | 4400MiB |
| GAT+AT | 50s | 1200MiB | 10min | 6000MiB | 15min | 12500MiB |
| GAT+FGAI | 60s | 1300MiB | 10min | 6000MiB | 20min | 16000MiB |
| Pro-GNN | 10min | 6000MiB | 10min | 8000MiB | - | OOM |
| GNN-SVD | 20min | 6000MiB | 20min | 8000MiB | - | OOM |
| GNNGuard | 25min | 1000MiB | 25min | 1000MiB | - | OOM |

it is evident that removing the adversarial loss component significantly degrades the F1 score performance of FGAI. However, there is a slight improvement in the g-JSD metric. On the other hand, omitting the top-$k$ loss component leads to a slight increase in the F1 score, surpassing even the performance with all components on GAT (when removing $L_1$). Nevertheless, there is a substantial drop in performance on g-JSD and g-TVD metrics. This indicates that during the training process, FGAI strikes a balance among the various components of the loss function, enabling the full method to achieve the best overall performance across all metrics. This observation underscores the high degree of joint effectiveness among the four terms within our objective function, collectively contributing to the enhancement of model prediction and explanation faithfulness. These findings reinforce the central roles played by the top-$k$ loss and TVD loss in improving the faithfulness of the model.

# I  Computation Cost Comparison

The usage of time and space is also a crucial metric for evaluating the excellence of a method, as it is related to the practical deployment and application in real-world scenarios. We test the time overhead and

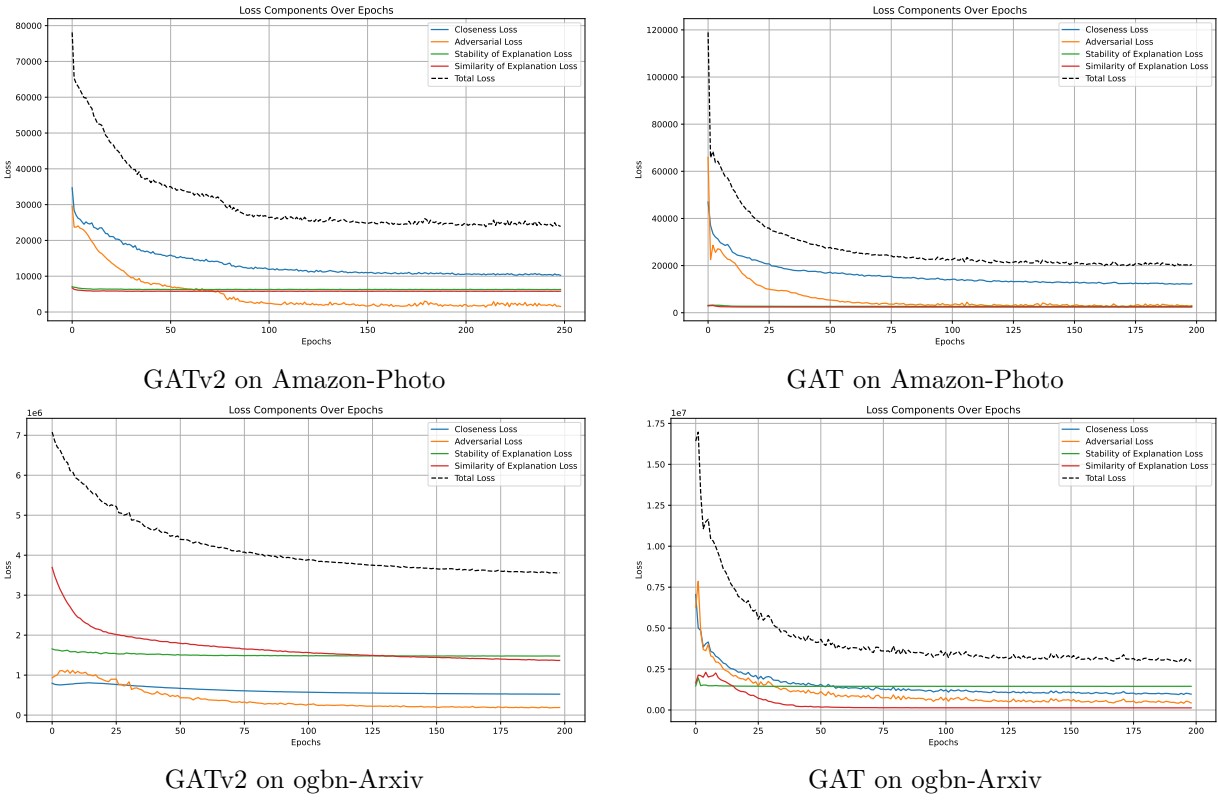

Figure 9: The illustration showing the training loss decreasing with increasing epochs for the GATv2 and GAT model applying FGAI on the Amazon-Photo and ogbn-Arxiv dataset.

GPU memory usage of some popular graph defense methods, namely Pro-GNN Jin et al. (2020), GNN-SVD Entezari et al. (2020) and GNNGuard Zhang & Zitnik (2020). The results are presented in Table 10. We observe that, compared to these methods, FGAI is a more efficient approach. To our knowledge, a primary reason for this discrepancy is that defense methods like GNN-SVD and GNNGuard require calculations on the dense form of the adjacency matrix of the graph. Consequently, this significantly limits their applicability on large-scale datasets. On ogbn-arXiv dataset (169,343 nodes, 1,166,243 edges), all these methods exceed the GPU memory limit (32GiB). In contrast, our method incurs relatively minimal additional overhead compared to the vanilla model, offering an efficient and cost-saving solution for the graph defense community.

## J   Limitations

This paper focuses solely on faithful interpretation for attention-based GNNs, such as GT or GAT. In the future, we aim to extend this work to all GNN variants, such as GCN, GraphSAGE, etc., to help drive research efforts across the entire community in the direction of GNN interpretability.

