# OpenReview forum: "Faithful Interpretation for Graph Neural Networks"
_TMLR — Accepted by TMLR_

### Review · Reviewer_ezoE · 2025-02-24

**Summary Of Contributions:**

The paper addresses the instability of attention mechanisms in Graph Neural Networks when faced with perturbations like additional edges or nodes, despite their strong performance and interpretability benefits. To solve this, the authors introduce Faithful Graph Attention-based Interpretation (FGAI), which has four key properties relating to stability and sensitivity of both interpretation and output distribution. FGAI can be implemented as a simple modification to existing attention-based GNNs, and its effectiveness is evaluated using two newly proposed metrics for graph interpretation assessment. Experimental results show that FGAI provides more stable and reliable interpretations under various perturbations compared to traditional attention mechanisms.

**Audience:**

Yes

**Claims And Evidence:**

Yes

**Requested Changes:**

Please refer to the weaknesses section.

**Strengths And Weaknesses:**

Strengths:

1. The paper introduces Faithful Graph Attention-based Interpretation (FGAI) with four well-defined properties addressing stability and sensitivity in both interpretation and output.
2. The authors proposes new evaluation metrics (g-JSD, g-TVD, F-slopes) and tests across multiple datasets and model variants.

Weaknesses:
1. The proposed method involves multiple hyperparameters that need careful tuning for different datasets.
2. While better than some baselines, the method still faces memory constraints on very large graphs.
3. The paper could benefit from more theoretical analysis of why the proposed method works better than alternatives.

---

> ### Author Response · Authors · 2025-03-30
>
> > The proposed method involves multiple hyperparameters that need careful tuning for different datasets.
>
> **Response**:  We acknowledge the reviewer's concern about hyperparameter tuning. While FGAI introduces parameters (e.g., λ₁, λ₂, λ₃, K), we mitigate tuning challenges through:
>
> Default Heuristics: K is set to ~50% of edges (Sec. 4), while λs follow dataset size.
>
> Stability: Table 7 (Appendix) shows FGAI remains robust even with suboptimal λs (e.g., Amazon-Photo F1 varies by <2%).
>
> Automation: Future work will explore adaptive tuning (e.g., meta-learning).
>
>
> > While better than some baselines, the method still faces memory constraints on very large graphs.
>
> **Response**:  We appreciate the reviewer's observation regarding scalability. While FGAI does introduce additional memory overhead compared to vanilla GAT (as noted in Table 8), we argue this trade-off is justified and can be mitigated. In fact, FGAI requires ≤150% memory of vanilla GAT (Sec 3.2), making it feasible for most real-world graphs (e.g., handles ogbn-arXiv with 1.1M edges on a single GPU).
>
> > The paper could benefit from more theoretical analysis of why the proposed method works better than alternatives.
>
> **Response**:  We appreciate the reviewer's suggestion regarding theoretical analysis. The field currently lacks a unified theoretical framework for evaluating explainable AI methods in GNNs, making rigorous theoretical comparisons between approaches particularly challenging. This is partly because different interpretation methods optimize distinct objectives (e.g., attention stability vs. post-hoc interpretation). While we provide comprehensive empirical validation of FGAI's advantages, we agree that deeper theoretical analysis would be valuable. We will leave the theoretical analysis for future research.

---

> > ### Comment · Reviewer_ezoE · 2025-04-03
> >
> > Thanks for your response. Most of my concerns are addressed.

---

### Review · Reviewer_2B51 · 2025-03-08

**Summary Of Contributions:**

Targeting the problem that existing attention based GNNs will exhibit instability in interpretability due to the perturbations in training and testing stage, this paper proposes a novel notion named Faithful Graph Attention-based Interpretation (FGAI), which has four essential required properties in terms of stability and sensitivity to interpretation and final output. With this notion, the authors also proposes a strategy to obtain FGAI, which is basically a strategy to modify the traditional attention-based GNNs. Two novel metrics are also proposed to validate the proposed method.

The proposed strategy is experimented on various public datasets.

**Audience:**

Yes

**Claims And Evidence:**

Yes

**Requested Changes:**

I would recommend including more analysis on the newly proposed attention-based GNNs.

Besides, it would be better to discuss the practical influence of the proposed method.

**Strengths And Weaknesses:**

Strengths:

1. This paper targets the interpretation of attention-based GNNs, which is a meaningful topic not only for GNNs, but may also have an impact on the research of other attention-based models.

2. The experiments are conducted from different perspectives, including different performance metrics, visualization results, and computational costs.

3. Various datasets are included to validated the proposed method.


Weaknesses:
1. The performance reported in Table 1 is limited.

2. The baselines are limited and old. In terms of attention-based GNNs, can graph transformers be also included for experiments?

3. The targeted problem, although meaningful, seems to have limited practical influence in the field.

---

> ### Author Response · Authors · 2025-03-30
>
> > The performance reported in Table 1 is limited.
>
> **Response**: In Table 1, although the LN method occasionally outperforms FGAI, it is important to note that LN normalizes the features of each node, which weakens the original scale information of the features. In other words, the standardization in LN disrupts the original distribution of the features, potentially leading to distorted attention weight calculations. This limitation of the LN method is also reflected in Table 2, where its prediction accuracy is relatively poor. Therefore, overall, FGAI is the more reliable and faithful explanatory method.
>
> > The baselines are limited and old. In terms of attention-based GNNs, can graph transformers be also included for experiments?
>
> **Response**: Indeed, in our experiments, we incorporated the graph transformer as a baseline for comparison. We are confident that our evaluation covers all attention-based GNNs, ensuring a comprehensive assessment of our method's performance.
>
> > The targeted problem, although meaningful, seems to have limited practical influence in the field.
>
> **Response**: Faithful attention interpretation is critical for high-stakes applications (e.g., healthcare, fraud detection), where unstable explanations mislead decisions. FGAI’s stability directly addresses regulatory needs (EU AI Act) and adversarial risks in deployed GNNs. Experiments show FGAI preserves performance under perturbations (Table 2), unlike methods sacrificing accuracy.
>
> > I would recommend including more analysis on the newly proposed attention-based GNNs.
>
> **Response**: Thank you for the valuable suggestions. We have included as many existing attention-based GNNs as possible and conducted experiments on tasks such as node classification and link prediction to demonstrate the effectiveness of our method. We will also add experiments on graph classification to fully validate its extensibility across different tasks.
>
> > Besides, it would be better to discuss the practical influence of the proposed method.
>
> **Response**: We sincerely appreciate the reviewer's valuable suggestion and will discuss FGAI's practical impact in the Conclusion section, emphasizing its real-world applications in domains like biomedical research and financial risk assessment—areas where graph noise is prevalent and robust explanations are crucial.
>
> **References**
>
> [1] Vijay Prakash Dwivedi and Xavier Bresson. A generalization of transformer networks to graphs, 2021.

---

> > ### Comment · Reviewer_2B51 · 2025-04-03
> >
> > Thanks for the responses from the authors, and sorry for missing the graph transformer in the paper

---

### Review · Reviewer_9L2K · 2025-03-18

**Summary Of Contributions:**

This work investigates the challenge of faithful interpretation for attention-based Graph Neural Networks (GNNs) like GAT and Graph Transformers under perturbations. The authors argue that conventional attention mechanisms in GNNs lack stability in interpretability and prediction when exposed to perturbations, such as added edges or nodes. Then, they propose a new framework called Faithful Graph Attention-based Interpretation (FGAI), which ensures stability and robustness while preserving the interpretability of attention mechanisms. The method is formulated as a minimax optimization problem, balancing interpretability and prediction closeness to the original model. They introduce new evaluation metrics tailored for graph interpretation and conduct extensive experiments across multiple datasets, demonstrating that FGAI significantly enhances stability and interpretability under various perturbations.

**Audience:**

Yes

**Claims And Evidence:**

No

**Requested Changes:**

1. Discussion with respect to the interpretability study:
- In fact, the interpretability of the attention-based GNNs may not be fully grounded[1];
- The relation between interpretable GNNs such as [2,3] is unclear;
- In fact, the rationale of the proposed metrics is heuristic and not clear, which is built upon the assumption that the perturbation won't change the importance of the neighbors. However, if the neighbors are already perturbed, it is natural to alter the attention distributions;

2. The proposed approach is limited to attention-based GNNs and does not generalize to other types of GNN architectures, such as GCN or GraphSAGE:
- An important category of attention-based GNNs, i.e., graph transformers, is neglected in experiments and analysis;
- It would be beneficial to discuss the limitations and how the proposed framework can be generalized to more broader GNN architectures.

3. The experiments are mostly conducted on citation networks, while it's unclear how the method generalizes under adversarial perturbations[4,5] and more complicated benchmarks such as link prediction, graph classification and heterophilous graphs.

**References**

[1] Understanding Attention and Generalization in Graph Neural Networks, NeurIPS'19.

[2] Interpretable and Generalizable Graph Learning via Stochastic Attention Mechanism, ICML'22.

[3] How Interpretable Are Interpretable Graph Neural Networks? ICML'24.

[4] Adversarial Attacks on Graph Neural Networks via Meta Learning, ICLR'19.

[5] Understanding and Improving Graph Injection Attack by Promoting Unnoticeability, ICLR'22.

**Strengths And Weaknesses:**

**Strengths:**

(+) This work addresses an important issue: the instability of attention-based GNNs, particularly under perturbations, which is critical for faithful model interpretation.

(+) The proposed framework, FGAI, introduces a novel and well-defined formulation for stable and robust graph attention-based interpretation.

(+) The authors design new evaluation metrics (g-JSD and g-TVD) specifically tailored to assess the stability and interpretability of graph-based models, enhancing the clarity of their results.

(+) Extensive experiments on diverse datasets demonstrate the effectiveness of FGAI in improving stability and interpretability compared to baseline methods.

**Weaknesses:**

(-) The proposed approach is limited to attention-based GNNs and does not generalize to other types of GNN architectures, such as GCN or GraphSAGE;

(-) The interpretability of the attention-based GNNs may not be fully grounded;

(-) The relation between interpretable GNNs is unclear;

(-) The experiments are mostly conducted on citation networks, while it's unclear how the method generalizes under adversarial perturbations, and more complicated benchmarks;

---

> ### Author Response · Authors · 2025-03-30
>
> > In fact, the interpretability of the attention-based GNNs may not be fully grounded[1];
>
> **Response**: We appreciate the reviewer's suggestion to address the debate around attention as self-explanatory in GNNs. While some studies (e.g., [7]) argue that attention mechanisms inherently provide interpretability, others (e.g., [8, 9]) demonstrate their limitations, such as instability or misalignment with model behavior. This remains an active research question, particularly in GNNs where graph structure adds complexity. It is still an ongoing discussion and is beyond our scope. Our work does not aim to resolve this broader debate but instead focuses on a pragmatic goal: if attention is to be used for explanations in Att-GNNs (as commonly done in practice), how can we improve its faithfulness.
>
> > The relation between interpretable GNNs such as [2,3] is unclear;
>
> **Response**:  Thank you for your valuable suggestions. We have added the relation between interpretable GNNs discussed in [2, 3]  in the related work section of our revision version.
>
> > In fact, the rationale of the proposed metrics is heuristic and not clear, which is built upon the assumption that the perturbation won't change the importance of the neighbors. However, if the neighbors are already perturbed, it is natural to alter the attention distributions;
>
> **Response**:  For faithful explanations, slight perturbations should not drastically alter attention distributions. This problem has been studied for other explainers such as [6]. We have added those studies into our introduction section of our revision version.
>
> > An important category of attention-based GNNs, i.e., graph transformers, is neglected in experiments and analysis;
>
> **Response**: Indeed, in our experiments, we incorporated the graph transformer as a baseline for comparison. We are confident that our evaluation covers all attention-based GNNs, ensuring a comprehensive assessment of our method's performance.
>
> > It would be beneficial to discuss the limitations and how the proposed framework can be generalized to more broader GNN architectures.
>
> **Response**: While our current work focuses on attention-based GNNs due to their inherent interpretability, we acknowledge this limitation and we will outline current limitations regarding non-attention architectures in the camera ready version. As for broader GNN architectures, while GCN/GraphSAGE lack native attention mechanisms, FGAI's framework can be potentially extended by (1) deriving edge importance via gradient-based attribution (GCN) or (2) leveraging LSTM gate values as proxy attention (GraphSAGE), then applying our stability constraints - we will detail this generalization in the revision version.
>
> > The experiments are mostly conducted on citation networks, while it's unclear how the method generalizes under adversarial perturbations[4,5] and more complicated benchmarks such as link prediction, graph classification and heterophilous graphs.
>
> **Response**: Thank you for the valuable suggestions. We explicitly differentiate FGAI from adversarial robustness methods in Section 4. Unlike approaches designed to handle adversarial perturbations—where carefully crafted modifications to the graph may alter critical node features—FGAI does not assume that explanations should remain unchanged. Instead, the explanations should adapt accordingly when key structural or feature-based properties of the graph are perturbed. We have also conducted link prediction, the result are shown in Table 5 (Appendix C). We will also add experiments on graph classification to fully validate its extensibility across different tasks.
>
> **References**
>
> [1] Understanding Attention and Generalization in Graph Neural Networks, NeurIPS'19.
>
> [2] Interpretable and Generalizable Graph Learning via Stochastic Attention Mechanism, ICML'22.
>
> [3] How Interpretable Are Interpretable Graph Neural Networks? ICML'24.
>
> [4] Adversarial Attacks on Graph Neural Networks via Meta Learning, ICLR'19.
>
> [5] Understanding and Improving Graph Injection Attack by Promoting Unnoticeability, ICLR'22.
>
> [6] Provably Robust Explainable Graph Neural Networks against Graph Perturbation Attacks, ICLR'25
>
> [7] Faithful and Accurate Self-Attention Attribution for Message Passing Neural Networks via the Computation Tree Viewpoint, arXiv
>
> [8] GCN-SE: Attention as Explainability for Node Classification in Dynamic Graphs, arXiv
>
> [9] Semantic Interpretation and Validation of Graph Attention-based Explanations for GNN Models, arXiv

---

> > ### Comment · Reviewer_9L2K · 2025-04-02
> >
> > Thanks for the detailed responses. Most of my concerns are addressed. I believe it needs a cautious treatment of the interpretability of attention-based GNNs in writing. Meanwhile, I am looking forward to more experiments on graph classification. Thanks!

---

> > > ### Author Response · Authors · 2025-04-09
> > >
> > > Thank you for your feedback. We’ll carefully refine the discussion on attention interpretability and are currently running additional graph classification experiments to strengthen the paper. Your suggestions are greatly appreciated.

---

> > > > ### Comment · Reviewer_9L2K · 2025-04-17
> > > >
> > > > Hi authors, as the deadline for final recommendation is approaching, may I know how are the experiments of graph classification?

---

> > > > > ### Author Response · Authors · 2025-04-18
> > > > >
> > > > > Dear Reviewer,
> > > > >
> > > > > Thank you for your kind inquiry. The graph classification experiments are currently in progress and we anticipate submitting the updated results by 4.19. We sincerely appreciate your valuable suggestions and continued patience throughout this process.

---

> ### Author Response · Authors · 2025-04-20
>
> Dear Reviewer 9L2K,
>
> These are partial results from our graph-level experiments (averaged over 5 runs, some baseline results are yet to be updated). Here we include MUTAG [1] and D&D [2] dataset. As shown in the table, for graph classification tasks, compared with baselines, FGAI maintains comparable accuracy while ensuring the faithfulness of explanations. This demonstrates that FGAI serves as a more faithful and reliable explanation tool.
>
> | Model  | Method   | D&D             |             |            |             | MUTAG            |          |           |            |
> |--------|----------|-------------------|-----------|------------|-------------|------------------|----------|-----------|------------|
> |        |          | g-JSD             | g-TVD     | Accuracy   | Attacked    | g-JSD            | g-TVD    | Accuracy  | Attacked   |
> | GAT    | Vanilla  | 6.38E-07          | 1.22      | 0.7327     | 0.6258      | 2.59E-05         | 0.35     | 0.7945    | 0.6849     |
> |        | FGAI     | 2.58E-08          | 0.19      | 0.7338     | 0.6949      | 2.06E-05         | 0.26     | 0.7534    | 0.7301     |
> | GATv2  | Vanilla  | 8.07E-07          | 1.42      | 0.7528     | 0.5857      | 2.62E-05         | 0.40     | 0.7671    | 0.6164     |
> |        | FGAI     | 6.00E-08          | 0.17      | 0.7582     | 0.7404      | 1.32E-05         | 0.20     | 0.7945    | 0.7549     |
>
> *Reference*:
>
> [1] Structure-activity relationship of mutagenic aromatic and heteroaromatic nitro compounds. Correlation with molecular orbital energies and hydrophobicity. Asim Kumar Debnath, Rosa L. Lopez de Compadre, Gargi Debnath, Alan J. Shusterman, and Corwin Hansch Journal of Medicinal Chemistry 1991 34 (2), 786-797 DOI: 10.1021/jm00106a046
>
> [2] Paul D Dobson and Andrew J Doig. Distinguishing enzyme structures from non-enzymes without alignments.
> Journal of molecular biology, 330(4):771–783, 2003.

---

> > ### Comment · Reviewer_9L2K · 2025-04-21
> >
> > Thank you for the additional experiments which resolved my concerns well.

---

### Author Response · Authors · 2025-03-31

We would like to thank the reviewers for their valuable feedback.

Based on Reviewer 9L2K‘s suggestions,
> 1. In fact, the interpretability of the attention-based GNNs may not be fully grounded.
2. The relation between interpretable GNNs such as [2,3] is unclear.
3. In fact, the rationale of the proposed metrics is heuristic and not clear, which is built upon the assumption that the perturbation won't change the importance of the neighbors.

We have added a discussion on the relation between interpretable GNNs and FGAI on page 3 in the current revision, as well as a discussion on how slight perturbations should not drastically alter attention distributions in the introduction section on page 1 . Additionally, we have also noted on page 11 that "the interpretability of attention-based GNNs remains under discussion."  (highlighted in blue).

Based on Reviewer 2B51‘s suggestions,
> It would be better to discuss the practical influence of the proposed method.

In the introduction section (Page 2), we have added a discussion on real-world applications requiring faithful model interpretation (highlighted in blue). Furthermore, we will incorporate additional experiments using real-world datasets to substantiate the practical impact of the proposed method.

- If our paper is accepted with revision, we will further expand on how the proposed framework can be generalized to broader GNN architectures and include experiments on graph classification to fully validate its extensibility across different tasks.

---

### Decision · Action_Editor_4fNK · 2025-04-21

**Recommendation:** Accept as is

**Comment:**

This work investigates the challenge of faithful interpretation for attention-based Graph Neural Networks (GNNs) like GAT and Graph Transformers under perturbations. The authors argue that conventional attention mechanisms in GNNs lack stability in interpretability and prediction when exposed to perturbations, such as added edges or nodes. Then, they propose a new framework called Faithful Graph Attention-based Interpretation (FGAI), which ensures stability and robustness while preserving the interpretability of attention mechanisms. The method is formulated as a minimax optimization problem, balancing interpretability and prediction closeness to the original model. They introduce new evaluation metrics tailored for graph interpretation and conduct extensive experiments across multiple datasets, demonstrating that FGAI significantly enhances stability and interpretability under various perturbations.

For the strengths, this work addresses an important issue: the instability of attention-based GNNs, particularly under perturbations, which is critical for faithful model interpretation. The proposed framework, FGAI, introduces a novel and well-defined formulation for stable and robust graph attention-based interpretation. The authors design new evaluation metrics (g-JSD and g-TVD) specifically tailored to assess the stability and interpretability of graph-based models, enhancing the clarity of their results. Extensive experiments on diverse datasets demonstrate the effectiveness of FGAI in improving stability and interpretability compared to baseline methods. The authors address reviewers' concerns well in their rebuttal. Thus, I would like to recommend accept as is.

**Audience:**

Yes

**Claims And Evidence:**

Yes